



# Development, intercomparison and evaluation of an improved mechanism for the oxidation of dimethyl sulfide in the UKCA model

Ben A. Cala[1,*], Scott Archer-Nicholls[1,#], James Weber[1,$], N. Luke Abraham[1,2], Paul T. Griffiths[1,2], Lorrie Jacob[1], Y. Matthew Shin[1], Laura E. Revell[3], Matthew Woodhouse[4], Alexander T. Archibald[1,2]

[1]Yusuf Hamied Department of Chemistry, University of Cambridge, Cambridge, CB2 1EW, UK
[2]National Centre for Atmospheric Science, Cambridge, CB2 1EW, UK.
[3]School of Physical and Chemical Sciences, University of Canterbury, Christchurch, New Zealand.
[4]CSIRO Oceans and Atmosphere, Aspendale, 3195, Australia.
*Now at Department of Ocean Systems (OCS), NIOZ Royal Netherlands Institute for Sea Research, Texel, the Netherlands
#Now at IT Services, University of Manchester, Manchester, M13 9PL, UK.
$Now at School of Biosciences, University of Sheffield, S10 2TN, UK.

*Correspondence to*: Alexander T. Archibald ata27@cam.ac.uk and Ben. A. Cala ben.cala@nioz.nl

**Abstract.** Dimethyl sulfide (DMS) is an important trace gas emitted from the ocean. The oxidation of DMS has long been recognised as being important for global climate through the role DMS plays in setting the sulfate aerosol background in the troposphere. However, the mechanisms in which DMS is oxidised are very complex and have proved elusive to accurately determine in spite of decades of research. As a result the representation of DMS oxidation in global chemistry-climate models is often greatly simplified.

Recent field observations, laboratory and ab initio studies have prompted renewed efforts in understanding the DMS oxidation mechanism, with implications for constraining the uncertainty in the oxidation mechanism of DMS as incorporated in global chemistry-climate models. Here we build on recent evidence and develop a new DMS mechanism for inclusion in the UKCA chemistry-climate model. We compare our new mechanism (CS2-HPMTF) to a number of existing mechanisms used in UKCA (including the highly simplified 3 reactions, 2 species, ST mechanism used in CMIP6 studies) and to a range of recently developed mechanisms reported in the literature through a series of global and box model experiments. Global model runs with the new mechanism enable us to simulate the global distribution of hydroperoxyl methyl thioformate (HPMTF), which we calculate to have a burden of 2.6-26 Gg S (in good agreement with the literature range of 0.7-18 Gg S). We show that the sinks of HPMTF dominate uncertainty in the budget, not the rate of the isomerisation reaction forming it, and that based on the observed DMS/HPMTF ratio, rapid cloud uptake of HPMTF worsens the model-observation comparison. Our box model experiments highlight that there is significant variance in simulated secondary oxidation products from DMS across mechanisms used in the literature, with significant divergence in the sensitivity of these products to temperature exhibited; especially for methane sulfonic acid (MSA). Our global model studies show that our updated DMS scheme performs better





than the current scheme used in UKCA when compared against a suite of surface and aircraft observations. However,
sensitivity studies underscore the need for further laboratory and observational constraints.
**1 Introduction**
It is estimated that 16-28 Tg S year$^{-1}$ are emitted in the form of dimethyl sulfide (DMS, $CH_3SCH_3$) from the ocean, making
DMS the most abundant biological source of sulfur in the Earth system (Andreae, 1990, Tesdal et al., 2015, Bock et al., 2021).
Elucidating the atmospheric fate of DMS has been a long standing goal of the atmospheric chemistry research community
owing to a proposed biogeochemical feedback cycle (CLAW; Charlson et al. 1987), whereby DMS oxidation is key to a
homeostatic feedback loop.  The initial steps in DMS oxidation are well understood (Barnes et al., 2006). Focusing on oxidation
via OH ($NO_3$), the most important oxidant during the daytime (nighttime), DMS is oxidised in the gas-phase through two main
pathways: the abstraction pathway forms the methylthiomethylperoxy radical (MTMP, $CH_3SCH_2OO$) in the first step, while
the addition pathway leads to dimethyl sulfoxide (DMSO, $CH_3SOCH_3$; and to a lesser extent $DMSO_2$) as  an important
intermediate.
DMS + OH/$NO_3 \rightarrow$ MTMP + $H_2O$/$HNO_3$ (abstraction)
DMS + OH $\rightarrow$ DMSO + $HO_2$ (addition)
Ultimately, the oxidation of DMS leads to products such as $H_2SO_4$ and sulfate ($SO_4^{2-}$), as these represent the highest oxidation
states of sulfur (S(VI)). Along the way from DMS, a number of secondary oxidation products such as sulfur dioxide ($SO_2$),
methane sulfonic acid (MSA, $CH_3SO_3H$) and carbonyl sulfide (OCS) can be formed, however the yields of these species
depend on the mechanisms involved, which themselves are a function of the chemical (e.g., levels of oxidants) and
environmental conditions (e.g., temperature and humidity). The yields of these products are relatively uncertain, with estimates
of the DMS-to-$SO_2$ yield spanning 14-96 % (von Glasow and Crutzen, 20114). The oxidation products can participate in
aerosol growth and in new particle formation, affecting the number of cloud condensation nuclei (CCN). As such DMS
oxidation can impact cloud formation and lifetime and hence climate; although the absolute effect is still highly uncertain due
to the uncertainty in DMS oxidation. Indeed, natural aerosols such as DMS contribute to large uncertainties in the radiative
forcing of the pre-industrial atmosphere (Carslaw et al., 2013).

Substantial discrepancies between different DMS oxidation mechanisms under different conditions have been found (de Bryn
et al., 2002; von Glasow and Crutzen, 2004). The intercomparison study by Karl et al. (2007) looked at six different chemistry
schemes in a box model and observed that $SO_2$ mixing ratios varied from 2 to 44 ppt. Differences between models are even
greater when looking at MSA yield (Karl et al., 2007, Hoffmann et al., 2021). The large uncertainties of product ratios indicate
the need for more observational constraints for DMS chemistry in models.





In the UK Chemistry and Aerosol model (UKCA) two different chemistry schemes are implemented: StratTrop (Archibald et
al., 2020), which is a simplified chemistry mechanism included in the UK Earth System Model (Sellar et al., 2019) and CRI-
Strat2 (Archer Nicholls et al., 2021; Weber et al., 2021). The DMS oxidation mechanism in StratTrop is, like those used in
many Earth System Models (ESMs), a very simple scheme. The StratTrop DMS mechanism only includes four reactions and
no intermediates for the DMS oxidation scheme.
(R1)  $DMS + OH \rightarrow SO_2 + MSA$
(R2)  $DMS + OH \rightarrow SO_2$
(R3)  $DMS + NO_3 \rightarrow SO_2$
(R4)  $DMS + O(^3P) \rightarrow SO_2$
Omitting intermediates might lead to a misrepresentation of the spatial distribution of oxidation products and an overestimation
in their formation since the intermediates might be subject to wet and dry deposition or cloud uptake. Because a unity yield of
$SO_2$ is assumed, a change in the distribution of oxidation products due to a changing climate cannot be evaluated.

CRI-Strat2 (hereafter CS2) (Archer-Nicholls et al., 2021, Weber et al., 2021) is a mechanism that aims to be of intermediate
complexity. CS2 includes 19 reactions and 7 intermediates (DMSO, MSIA, MTMP, $CH_3S$, $CH_3SO$, $CH_3SO_2$, $CH_3SO_3$) as part
of its DMS scheme and is primarily based on the work of von Glasow and Crutzen (2004). Whilst the CS2 DMS mechanism
is much more complex than the StratTrop scheme, it represents an understanding of DMS chemistry that is far from up-to-
date.

In this work, the gas-phase DMS oxidation by OH and $NO_3$ in CS2 is updated according to the current scientific understanding.
The greatest update is the inclusion of the recently discovered intermediate hydroperoxymethyl thioformate (HPMTF,
$HOOCH_2SCHO$), which is formed through the autoxidation of the methylthiomethyl peroxy radical (MTMP, $CH_3SCH_2OO$)
in the abstraction pathway (Wu et al., 2015, Berndt et al., 2019, Veres et al. 2020). Currently, it is estimated that 30-40% of
DMS yields HPMTF (Veres et al., 2020). There are large uncertainties about the value of $k_{isom,1}$, the rate constant of the first
H-shift, which is the rate determining step for HPMTF formation (**Figure 1**). This determines if autoxidation can compete with
or surpass the bimolecular reactions of MTMP with $HO_2$ and NO. The chamber study by Ye et al. (2021) estimates a probability
distribution based on their measurements with one geometric standard deviation spanning an order of magnitude. The
isomerization rate constant is predicted using *ab initio* methods to be strongly temperature dependent, indicating that  this
pathway could be more relevant under a warming climate (Wu et al., 2015; Veres et al., 2020).

As of now, the fate of HPMTF in the atmosphere is largely unknown. Wu et al. (2015) postulate further oxidation with OH,
ultimately yielding $SO_2$ as the dominant product and OCS as a side product. Veres et al. (2020) observe an abrupt decrease of
HPMTF mixing ratio in clouds and therefore suggest that heterogeneous loss to aerosol and cloud uptake plays a big role.
Vermeuel et al. (2020) support this hypothesis: they find a diurnal profile of HPMTF in the vicinity of California's coast and





suggest this is due to the consistent diurnal profile of cloud present. This hypothesis is further supported by the study by Novak
et al. (2021), which looks at two case studies and concludes that cloud uptake determines the lifetime of HPMTF. Novak et
al. (2021) found that cloud-uptake of HPMTF reduces $SO_2$ production from DMS by over a third, while providing a more
direct pathway to sulfate formation. On the contrary, the chamber study and calculation of Henry's law constant by Wollesen
de Jonge et al. (2021) predict that HPMTF does not directly contribute to new particle formation or aerosol growth. Instead,
their study proposes aqueous oxidation by OH, ultimately still yielding gas-phase $SO_2$. Khan et al. (2021) stress the importance
of photolysis as a potential loss pathway, which might explain the observed diurnal concentrations throughout the day. Overall,
loss processes of HPMTF are poorly understood.

In this work, we perform a series of updates to the CS2 DMS oxidation scheme which are evaluated against the current CS2
and the very simplified DMS chemistry in StratTrop. The aim of this work is to improve the representation of DMS chemistry
in UKCA and determine the influence of some of the major mechanistic uncertainties on model simulated $SO_2$ levels compared
against ATom observations (Wofsy et al., 2018; Veres et al., 2020). Our study includes a comprehensive set of box model
studies, including an intercomparison of our new DMS scheme against other recently reported schemes in the literature, and
global 3D simulations with the UKCA model. Sensitivity studies with slower loss, a faster production, and cloud and aerosol
uptake of HPMTF are performed to investigate the effects of the uncertainty in HPMTF formation and depletion on the
distribution and burden of $SO_2$ and sulfate (given their importance in climate).

## 2 Methods

### 2.1 Model description

#### 2.1.1 Set up

**Box model**

For the box model experiments, BOXMOX (Knote et al., 2015), the box modelling extension to the Kinetic PreProcessor
(KPP) (Sandu and Sander, 2006) was used. The initial and background concentrations of the species were set to be
representative of the remote marine boundary layer (MBL) (and are detailed in **Table S1**). $NO_x$ concentration was kept at
approximately 10 ppt, unless otherwise specified.
The box model set up simulates an MBL air parcel exchanging with the free troposphere. The diurnal profile of the planetary
boundary layer height was modelled after the diurnal profile of the MBL in Ho et al. (2015) (**Table S2**). Mixing of the air
within the box with the free troposphere is described by the increases of box height: it is assumed that changes in the box
volume are due to the influx of background air. Emissions of DMS are added at $3.48 \times 10^9$ molec. $cm^{-2}$ $s^{-1}$ (consistent with
von Glasow and Crutzen, 2004). Emissions mix instantaneously within the box. Temperature varies throughout a 24 hour





period between 289 - 297 K, with a mean of 293 K (**Table S2**). Photolysis reactions are scaled depending on the time of day,
and make use of the pre-calculated "J" rates obtained from the MCMv3.3.1. The simulations were run for 192 hours (8 days)
with 10 minute time steps. CRI v2.2 R5 (CS2) (Jenkin et al., 2019; Weber et al., 2021) was employed as the base chemical
mechanism. Unless otherwise specified, only reactions of the DMS scheme were changed. Neither dry nor wet deposition was
included in the box model experiments. The analysis of the BOXMOX simulations discussed in Section 3.1.1 and 3.2.1 focuses
on the continuous (hourly) output. In Section 3.1.2 and 3.2.2, simulations with a prescribed temperature (260 - 310 K, step
size: 5 K) were conducted. The data from day 7 and 8 of the runs was averaged to enable the effects of changes in the
temperature on species concentration simulated in the box model to be calculated (following Archibald et al., 2010)
**3D simulations**
For the 3D simulations we use UKCA, the chemistry and aerosol component of UKESM1, 1.25°×1.875° with 85 vertical levels
up to 85 km (Walters et al., 2019), and the GLOMAP-mode aerosol scheme, which simulates sulfate, sea salt, black carbon
(BC), organic matter, and dust but does not simulate currently nitrate aerosol (Mulcahy et al., 2020). Simulations were run for
18 months, using the first 6 months as spin up. In order to look at high time resolution output simulations were re-run for
limited time periods using the re-start files of the longer runs but outputting data at hourly frequency.
Temperature and horizontal wind fields were nudged (Telford et al., 2013) in all model runs to the Era-Interim atmospheric
reanalysis from ECMWF (Dee et al., 2011). This constrains the different simulations to consistent meteorology, thus
preventing differences in meteorology complicating the attribution of differences resulting from the chemical mechanism
changes, and replicating the atmospheric conditions experienced when the observations were recorded as closely as possible.
Nudging only occurred above ∼1200 m in altitude, and thus the majority of the planetary boundary layer was not nudged. The
model runs were atmosphere-only runs with prescribed sea surface temperatures (SSTs). $CO_2$ is not emitted but set to a constant
field, while methane, CFCs, and $N_2O$ are prescribed with constant lower boundary conditions, all at 2014 levels (Archibald et
al., 2020).

The emissions used in this study for UKCA are the same as those from Archer-Nicholls et al (2021) and are those developed
for the Coupled-Model Intercomparison Project 6 (CMIP6) (Collins et al., 2017). Anthropogenic and biomass burning
emissions data (including DMS) for CMIP6 are from the Community Emissions Data System (CEDS), as described by Hoesly
et al. (2018). All runs used time slice 2014 emissions for anthropogenic and biomass burning emissions. Oceanic emissions of
CO, $C_2H_4$. $C_2H_6$, $C_3H_6$ and $C_3H_8$ were from the POET 1990 data set (Olivier et al., 2003), and all terrestrial biogenic emissions
except isoprene and monoterpenes were based on 2001–2010 climatologies from Model of Emissions of Gases and Aerosols
from Nature under the Monitoring Atmospheric Composition and Climate project (MEGAN-MACC) (MEGAN) version 2.1
(Guenther et al., 2012). Emissions of isoprene and monoterpenes were simulated by the interactive biogenic volatile organic
compound (iBVOC) emissions system (Pacifico et al.. 2011), the standard approach for UKESM1's contributions to CMIP6





(Sellar et al., 2019). Emissions of isoprene and monoterpenes are calculated interactively based on temperature, $CO_2$,
photosynthetic activity and plant functional types for each grid cell. Oceanic emissions of DMS are calculated from seawater
DMS concentrations (Sellar et al., 2019). In the atmosphere-only setup employed here seawater DMS concentrations for 2014
from a UKESM1 fully-coupled SSP3-70 ensemble member were prescribed. The DMS emission flux from the ocean used in
the model was 16 Tg S $yr^{-1}$ and therefore on the low end of estimates of oceanic DMS emissions (e.g., Lana et al., 2011; Bock
et al., 2021).

While the StratTrop mechanism and the variants of the CS2 mechanism all use the same raw emissions data, the additional
emitted species required by CS2 means the total mass of emitted organic compounds is greater in CS2, and the lumping of
species for emissions is also different. The approach and consequences are discussed in Archer-Nicholls et al (2021).

**2.1.2 Model runs**

**Table 1:** Configuration of model runs in this study. The last two columns indicate whether this scheme was used for the
BOXMOX experiments or the UKCA runs or both. Additional BOXMOX simulations were performed and the results of which
are included in the Supplementary Information (SI) for completeness.

| Alias | Description | Used for: BOXMOX | UKCA |
|---|---|---|---|
| CS2 | Base simulation, standard CRIStrat2 (or CRIv2.2R5) scheme | ✓ | ✓ |
| ST | StratTrop chemistry scheme <br> *(ST - CS2 = ΔST; change between ST and CS2)* | ✓ | ✓ |
| ST~CS2 | StratTrop DMS scheme but CS2 oxidants <br> *(ST~CS2 - CS2 = ΔCC; change between CS2 and the ST DMS scheme only)* | ✓ | - |
| CS2-HPMTF | CS2 + updates in **Table 2** and **Table 3** <br> *(CS2-HPMTF - CS2 = ΔUPD; effects of all updated made to the scheme)* | ✓ | ✓ |
| CS2-UPD-DMS | CS2 + updates in **Table 2** = CS2-HPMTF - updates in **Table 3** <br> *(CS2-HPMTF - CS2-UPD-DMS = ΔHPMTF; effects of the isom. Pathway only)* | ✓ | - |
| CS2-HPMTF-CLD | CS2-HPMTF + cloud and aerosol uptake (γ = 0.01) <br> *(CS2-HPMTF-CLD - CS2-HPMTF = ΔCLD; gives the effects of cloud and aerosol uptake of HPMTF)* | - | ✓ |
| CS2-HPMTF-FL | CS2-HPMTF + faster total loss of HPMTF to OH ($5.5 \times 10^{-11}$ $s^{-1}$) <br> *(CS2-HPMTF-FL - CS2-HPMTF = ΔFL; gives the effects of faster gas phase loss of HPMTF)* | SI | ✓ |
| CS2-HPMTF-FP | CS2-HPMTF + isomerisation *A*-factor scaled by a factor of 5, see Wollesen de Jonge et al. (2021)) | SI | ✓ |





*(CS2-HPMTF-FP - CS2-HPMTF = ΔFP; gives the effects of faster HPMTF*

*production)*


Simulations are performed with the standard or updated DMS scheme to quantify the impacts of the mechanistic changes.
Details are given in **Table 1**. We chose as our base run a simulation with the CRIStrat2 chemistry scheme hereafter referred
to as CS2 (Weber et al., 2021). We perform two simulations with StratTrop (hereafter ST): ST is the default mechanism as
used in UKESM1, while ST~CS2 uses the ST DMS chemistry (R1-R4) but all other reactions ($HO_x$, $NO_x$, VOC etc) are
identical to CS2. This allows us to attribute the changes arising solely to differences in the oxidising capacity/environment
(driven by the chemistry not strongly coupled to DMS) and isolate the role due to differences in the DMS reactions themselves.

In updating the representation of DMS chemistry for UKCA a number of changes were considered. Broadly these fall into two
categories: 1) Incorporation of the chemistry of HPMTF (shown in red in **Figure 1**) 2) updates to other aspects of DMS
oxidation chemistry (shown in blue in **Figure 1**). CS2-HPMTF is used to identify the fully updated DMS mechanism (**Table
2**, **Table 3**). All other runs act as sensitivity runs. CS2-UPD-DMS allows the evaluation of only updating the standard DMS
chemistry (**Table 2**), without the addition of the isomerization branch and HPMTF formation (**Table 3**). CS2-HPMTF-CLD
adds cloud and aerosol uptake of HPMTF with subsequent sulfate formation, similar to Novak et al. (2021). With CS2-
HPMTF-FP and CS2-HPMTF-FL the effects of faster production and faster loss of HPMTF can be assessed.








**Figure 1.** Schematic summary of the changes and additions to the gas-phase DMS oxidation mechanism in CS2. The current chemistry in CS2 is in black, changes associated with CS2-UPD-DMS are shown in blue and changes associated with the addition of the isomerization pathway for CS2-HPMTF in red.





## 2.2 New mechanism development

The current CS2 DMS oxidation mechanism is based on von Glasow and Crutzen (2004). This mechanism is based on an outdated understanding of DMS oxidation, which excludes key pathways and intermediates that are now known to be well established (Barnes et al., 2006) as well as more recent pathways and products that have been shown to be important (Veres et al., 2020). Our aim with the development of the new mechanism is to build upon the existing mechanism in CS2 and to update and extend it. To this end we performed a literature review and constructed a number of mechanistic variants that were examined in a series of box model experiments. As with all mechanism development exercises a series of target compounds were chosen to reduce the mechanism to achieve a scheme that is parsimonious; for use in a 3D chemistry-climate model. In our study we chose DMS, $SO_2$, sulfate and HPMTF as the key target molecules for mechanism optimization. **Figure 1** shows the two-step improvement of this mechanism. First, the improvement of the standard chemistry by updating rate constants for existing reactions in the scheme or the addition of reactions that were missing (denoted with blue colouring in **Figure 1**), and second, the addition of the HPMTF pathway (in red in **Figure 1**). The focus in this study is on gas-phase DMS oxidation by OH and $NO_3$. While other studies include DMS oxidation by BrO and Cl, the contribution is either negligible or there is a large uncertainty attached due to substantial discrepancies between/within models and measurements of halogens and halogen oxides (Wang et al., 2021; Fung et al., 2022).

## 2.2.1 Updating the standard DMS chemistry in CRIStrat 2

The H-abstraction pathway (reaction 1a,b) generates MTMP which is then further oxidised to $SO_2$ or $CH_3SO_2$ (reactions 2-7). The OH-addition pathway (reaction 1c) leads to dimethyl sulfoxide (DMSO, $(CH_3)_2SO$) and methanesulfinic acid (MSIA, $CH_3S(O)OH$) (reactions 8,9) and further oxidation through to $CH_3SO_2$ (reactions 10-12). Both pathways and the changes made are summarised in **Table 2**. The newly added reactions and their respective rate constants are largely based on Atkinson et al. (2004), the MCMv3.3.1 (Jenkin et al. 2015), and the primary literature therein.

The oxidation of MTMP by $HO_2$ (reaction 2c) was not previously included in the CS2 mechanism, but is expected to play a significant role at the low $NO_x$ conditions over the remote ocean. Based on other $RO_2 + HO_2$ reactions, $CH_3SCH_2OOH$ is the expected product, which has been detected through mass spectroscopy (Butkovskaya and LeBras, 1994). Since no experimental measurements exist for the kinetics of this reaction, the rate constant provided in the MCM was used. It is based on a generic expression, defined on the basis of available room temperature and temperature dependent data for alkyl and β-hydroxy $RO_2$ and it is dependent on the number of carbon atoms. Further oxidation of $CH_3SCH_2OOH$ leads to the formation of methylthiolformate (MTF, $CH_3SCHO$) (reaction 3), a species that has been detected in chamber studies before under low $NO_x$ conditions (Arsene et al., 1999, Urbanski et al., 1998). MTF decomposes to $CH_3S$ (reaction 4), an intermediate that is already part of the CS2 DMS scheme as a reaction product of MTMP (reaction 2a,b).






CH$_3$S can add an O$_2$ to form a weakly bound adduct, CH$_3$SOO (reaction 5c). At 298 K at sea level, approximately one-
third of CH$_3$S is present as the CH$_3$SOO adduct and at colder temperatures this ratio is even greater (75% at 273 K)
(Turnipseed et al., 1992). CH$_3$SOO can decompose to CH$_3$ and SO$_2$ (reaction 6a), which proceeds through isomerization
to CH$_3$SO$_2$, followed by rapid thermal decomposition (McKee, 1993, Butkovskaya and Barnes, 2002, Chen et al. 2021).
Previous modelling studies, such as Hoffmann et al. (2016), include the isomerization step forming CH$_3$SO$_2$ but omit the
decomposition. This could lead to a higher yield of MSA in those studies.

CH$_3$S can also be oxidised by O$_3$ and NO$_3$ to CH$_3$SO (reaction 5a,b). Measurements by Borissenko et al. (2003) show that
O$_3$ oxidation of CH$_3$SO results in a 100% yield of SO$_2$ at pressures over 500 Torr (0.6 bar). Since the pressure in the marine
boundary layer where most of DMS oxidation takes place is above this threshold, the products of reaction 7c_old were
updated accordingly (reaction 7c). Additionally, the branching ratios of CH$_3$SO oxidation by NO$_3$ to CH$_3$SO$_2$ and SO$_2$
were revised to also match the findings by Borissenko et al. (2003).

While some CH$_3$SO$_2$ stems from the NO$_3$ oxidation of CH$_3$SO, it is mainly formed through oxidation of MSIA (reaction
9a,c), especially under low NO$_x$ conditions. CH$_3$SO$_2$ can decompose to SO$_2$ (reaction 10a) or be oxidised further by O$_3$ or
NO$_3$ to CH$_3$SO$_3$ (reaction 10b,c). CH$_3$SO$_3$ itself can react to form MSA (reaction 11a). CH$_3$SO$_3$ can also decompose to
SO$_3$, similar to the decomposition reaction of CH$_3$SO$_2$, although it is assumed that this reaction is more endothermic
(Barone et al., 1995). The rate constant cited by von Glasow and Crutzen (2004) that was previously implemented in CS2,
could not be found in the cited primary literature (reaction 11b_old). Here, the rate constant of the decomposition reaction
was updated to the rate constant used in the MCMv3.3.1, which is — as for the decomposition of CH$_3$SO$_2$ — based on
Barone et al. (1995). We note that a more recent study, by Cao et al. (2013), calculates the rate constant for the thermal
decomposition of CH$_3$SO$_3$ to be 12 s$^{-1}$; a factor of 80 larger than the value adopted here based on the MCMv3.3.1.

MSA is formed either through oxidation of MSIA (reaction 9b) or through the reaction of HO$_2$ with CH$_3$SO$_3$ (reaction
11a). The default configuration of UKCA (for example as run in UKESM1) does not include any sinks for MSA and it is
not treated as a species, which prevents the comparison of MSA concentrations with observational results. Here, wet
deposition of MSA is added with a Henry's law coefficient of $1 \times 10^9$ M atm$^{-1}$ (Campolongo et al., 1999). We note that
Wollesen de Jonge et al. (2021) calculated the Henry's law coefficient to be two magnitudes lower and so this might be
an overestimate. Dry deposition for MSA is added based on the implemented values for HCOOH in CRI. Additionally,
the gas-phase oxidation of MSA by OH is added. Barnes et al. (2006) suggest this pathway is expected to play a minor
role.





Wet deposition was added for MSIA with Henry's law constant of $1\times10^8$ M atm$^{-1}$ (Barnes et al., 2006). Dry deposition is

omitted for DMSO and MSIA since they are expected to be relatively short-lived.

**Table 2:** Summary of the H-abstraction and OH-addition branches in the DMS oxidation pathway. Reactions in **bold** are

newly added in this work.

| No. | Reactions | Rate (cm$^3$ molecule$^{-1}$ s$^{-1}$) | Reference |
|---|---|---|---|
| 1a | DMS + OH → MTMP + H$_2$O | $1.12\times10^{-11}$ exp$^{(-250/T)}$ | IUPAC SOx22 (upd. 2006) |
| 1b | DMS + NO$_3$ → MTMP + HNO$_3$ | $1.90\times10^{-13}$ exp$^{(520/T)}$ | Atkinson et al. (2004) |
| 1c | DMS + OH → DMSO + HO$_2$ | see note[a] | IUPAC SOx22 (upd. 2006) |
| 2a | MTMP + NO → HCHO + CH3S + NO2 | $4.90\times10^{-12}$ exp$^{(263/T)}$ | von Glasow and Crutzen (2004) |
| 2b | MTMP + MTMP → 2 HCHO + 2 CH$_3$S | $1.0\times10^{-11}$ | von Glasow and Crutzen (2004) |
| **2c** | **MTMP + HO$_2$ → CH$_2$SCH$_2$OOH** | **$2.91\times10^{-13}$ exp$^{(1300/T)}$ × 0.387** | MCMv3.3.1 |
| **3** | **CH$_2$SCH$_2$OOH + OH → CH$_3$SCHO** | **$7.03\times10^{-11}$** | MCMv3.3.1 |
| **4** | **CH$_3$SCHO + OH → CH$_3$S + CO** | **$1.11\times10^{-11}$** | MCMv3.3.1 |
| 5a | CH$_3$S + O$_3$ → CH$_3$SO | $1.15\times10^{-12}$ exp$^{(432/T)}$ | Atkinson et al. (2004) |
| 5b | CH$_3$S + NO$_2$ → CH$_3$SO + NO | $3.00\times10^{-12}$ exp$^{(210/T)}$ | Atkinson et al. (2004) |
| **5c** | **CH$_3$S + O$_2$ → CH$_3$SOO** | **$1.20\times10^{-16}$ exp$^{(1580/T)}$ × [O$_2$]** | Atkinson et al. (2004) |
| **6a** | **CH$_3$SOO → CH$_3$O$_2$ + SO$_2$** | **$5.60\times10^{+16}$ exp$^{(-10870/T)}$** | Atkinson et al. (2004) |
| **6b** | **CH$_3$SOO → CH$_3$S + O$_2$** | **$3.50\times10^{+10}$ exp$^{(-3560/T)}$** | MCMv3.3.1 (based on: McKee (1993), and Butkovskaya and Barnes (2002)) |
| 7a | CH$_3$SO + NO$_2$ → CH$_3$SO$_2$ + NO | $1.2\times10^{-11}$ × **0.75** | Borrisenko et al. (2003), Atkinson et al. (2004) |
| 7b | CH$_3$SO + NO$_2$ → SO$_2$ + CH$_3$O$_2$ + NO | $1.2\times10^{-11}$ × **0.25** | Borrisenko et al. (2003), Atkinson et al. (2004) |
| 7c_old | CH$_3$SO + O$_3$ → CH$_3$SO$_2$ | $6.0\times10^{-13}$ | Von Glasow and Crutzen (2004) |
| **7c** | **CH$_3$SO + O$_3$ → CH$_3$O$_2$ + SO$_2$** | **$4\times10^{-13}$** | Borrisenko et al. (2003), IUPAC SOx61 (upd. 2006) |
| 8 | DMSO + OH → MSIA + CH$_3$O$_2$ | $8.7\times10^{-11}$ × 0.95 | von Glasow and Crutzen (2004) |
| 9a | MSIA + OH → CH$_3$SO$_2$ + H$_2$O | $9.0\times10^{-11}$ × 0.95 | von Glasow and Crutzen (2004) |
| 9b | MSIA + OH → MSA + HO$_2$ + H$_2$O | $9.0\times10^{-11}$ × 0.05 | von Glasow and Crutzen (2004) |
| 9c | MSIA + NO$_3$ → CH$_3$SO$_2$ + HNO$_3$ | $1.0\times10^{-13}$ | von Glasow and Crutzen (2004) |
| 10a | CH$_3$SO$_2$ → CH$_3$O$_2$ + SO$_2$ | $5.0\times10^{-13}$ exp$^{(-9673/T)}$ | MCMv3.3.1 (based on: Barone et al. (1995)) |
| 10b | CH$_3$SO$_2$ + O$_3$ → CH$_3$SO$_3$ | $3.0\times10^{-13}$ | von Glasow and Crutzen (2004) |
| 10c | CH$_3$SO$_2$ + NO$_2$ → CH$_3$SO$_3$ + NO | $2.2\times10^{-12}$ | Atkinson et al. (2004) |
| 11a | CH$_3$SO$_3$ + HO$_2$ → MSA | $5.0\times10^{-11}$ | von Glasow and Crutzen (2004) |
| 11b_old | CH$_3$SO$_3$ → CH$_3$O$_2$ + H$_2$SO$_4$ | $1.36\times10^{14}$ exp$^{(-11071/T)}$ | von Glasow and Crutzen (2004) |
| **11b** | **CH$_3$SO$_3$ → CH$_3$O$_2$ + SO$_3$** | **$5.0\times10^{13}$ exp$^{(-9946/T)}$** | MCMv3.3.1 (based on: Barone et al. (1995)) |





| 12 | **MSA + OH → CH₃SO₃** | **$2.24×10^{-14}$** | MCMv3.3.1 |

[a] $9.5×10^{-39}\exp^{(5270/T)}×[O_2] / (1+7.5×10^{-29}\exp^{(5610/T)}×[O_2])$

### 2.2.2 The addition of the isomerization branch

Following the discovery of HPMTF (Veres et al., 2020) the pathway forming this molecule has now been well established (Wu et al., 2015; Veres et al., 2020; Berndt et al., 2019; Ye et al., 2021). The reactions of the isomerization branch that were added to CS2 (summarised in **Figure 1** and **Table 3**) were identified as those most important in determining $SO_2$ and HPMTF concentrations through sensitivity studies conducted using our box model setup. Details of these box model sensitivity studies (and the discarded reaction pathways that were found to not be significant) are included in the supplement. In this sense, species like $HOOCH_2SCH_2OOH$, included in the studies by Khan et al. (2021) were neglected from our mechanism as this was found to have minor impact on the $SO_2$ and HPMTF simulated in the box model experiments. The reactions that were added include the autoxidation of MTMP to HPMTF in one step (reaction 2c) and the oxidation of HPMTF by OH, forming OCS (reaction 13b) and $HOOCH_2S$ (reaction 13a) with further oxidation to $SO_2$ (reactions 14-16). The equilibrium with the $O_2$-adduct, $HOOCH_2SOO$, and its subsequent decomposition (reaction 14c, 15a,b) was included with kinetics equivalent to $CH_3SOO$ (reaction 5c, 6a,b). Photolysis was found to be a minor pathway of HPMTF loss in our marine boundary layer box model setup ($< 10\%$) and was omitted from the final mechanism used here; contrary to the importance of photolysis of HPMTF found by Khan et al. (2021).

Dry deposition of HPMTF is set using the same parameters in UKCA as other soluble gas-phase compounds, such as $CH_3OOH$ and $H_2O_2$, which yield an average deposition velocity similar to the observations of Vermeuel et al. (2020) of 0.75 cms$^{-1}$. For wet deposition of HPMTF, the Henry's law coefficient calculated by Wollesen de Jonge et al. (2021) was used.

For the sensitivity runs described in **Table 1**, some changes are made to the values in **Table 3**. In DMS-HPMTF-FP, the rate constant of reaction 2d is scaled by a factor of 5.0: Berndt et al. (2019) experimentally determined the rate constant at 295 K as 0.23 s$^{-1}$. Here the A-factor is scaled to match this value, while keeping the temperature dependence calculated by Veres et al. (2020) (following Wollesen de Jonge et al. (2021)). DMS-HPMTF-FL uses a rate constant 5.5 times faster for the total loss of HPMTF to OH (reaction 13a,b), which was recommended as an upper bound by Vermeuel et al. (2020) and following Khan et al. (2021). In the remaining sensitivity run CS2-HPMTF-CLD, heterogeneous uptake to both clouds and aerosols was added with reactive uptake coefficient (γ) of 0.01 (following Novak et al., 2021).

**Table 3:** Summary of the isomerization branch of the H-abstraction pathway.

| No. | Reaction | Rate (cm³ molecule⁻¹ s⁻¹) | Reference |
|-----|----------|---------------------------|-----------|
| 2d | MTMP → HPMTF + OH | see note[a] | Veres et al. (2020) |





| | | | |
|---|---|---|---|
| 13a | $HPMTF + OH \rightarrow HOOCH_2S + H_2O + CO$ | $1.0\times10^{-11} \times 0.9$ | this work |
| 13b | $HPMTF + OH \rightarrow OCS + OH + HCHO + H_2O$ | $1.0\times10^{-11} \times 0.1$ | this work |
| 14a | $HOOCH_2S + O_3 \rightarrow HOOCH_2SO$ | $1.15\times10^{-12} \exp^{(430/T)}$ | Wu et al. (2015) |
| 14b | $HOOCH_2S + NO_2 \rightarrow HOOCH_2SO + NO$ | $6.00\times10^{-11} \exp^{(240/T)}$ | Wu et al. (2015) |
| 14c | $HOOCH_2S + O_2 \rightarrow HOOCH_2SOO$ | $1.20\times10^{-16} \exp^{(1580/T)} \times [O_2]$ | this work |
| 15a | $HOOCH_2SOO \rightarrow HOOCH_2S + O_2$ | $3.50\times10^{+10} \exp^{(-3560/T)}$ | this work |
| 15b | $HOOCH_2SOO \rightarrow HCHO + OH + SO_2$ | $5.60\times10^{-16} \exp^{(-10870/T)}$ | this work |
| 16a | $HOOCH_2SO + O_3 \rightarrow HCHO + OH + SO_2$ | $4\times10^{-13}$ | Wu et al. (2015) |
| 16b | $HOOCH_2SO + NO_2 \rightarrow HCHO + OH + NO + SO_2$ | $1.2\times10^{-11}$ | Wu et al. (2015) |

[a] $2.24\times10^{+11} \exp^{(-9800/T)} \exp^{(1.03e8/(T\times T\times T))}$

## 2.3 Description of observational data

### 2.3.1 The NASA Atmospheric Tomography (ATom) mission

An observational dataset used to compare with the model simulations stems from the fourth flight campaign of the NASA Atmospheric Tomography mission (ATom-4). ATom-4 took place during April and May 2018, and completed a global circuit around the Americas: from the Arctic to the Antarctic over the remote Pacific and Atlantic Ocean at varying altitudes up to 12 km. A vast number of atmospheric species were measured, including DMS, HPMTF, and $SO_2$ (Wofsy et al., 2018).

In order to compare the 3D model outputs with the data from the ATom-4 campaign, the hourly outputs from the respective model runs were interpolated in regards to time and space to generate the data along the flight path. Only model data at times where valid atmospheric measurements were available are taken into account, resulting in 313 data points for DMS (Whole Air Sampling) and 36,652 for $SO_2$ (Laser Induced Fluorescence).

### 2.3.2 Surface observations

Other observational measurements are monthly averages (mean) from the years 1990 to 1999 for DMS measurements made on Amsterdam Island (37°S, 77°E) in the southern Indian Ocean (Sciare et al., 2000) and the monthly means from 1991 to 1995 for sulfate at the Dumont d'Urville station (66°S, 140°E) at the coast of Antarctica (Minikin et al., 1998). The diel profile of HPMTF as measured at Scripps Pier in July 2018 was taken from Vermeuel et al. (2020)



## 3 Comparison of DMS oxidation pathways (BOXMOX)

Here we present the results of a series of box model simulations using the BOXMOX model (Weber et al., 2020). With BOXMOX we look at the diversity in results from simulations using a range of mechanisms, including our newly developed mechanism. These simulations are not constrained to observations or simulation chamber data. The set-up of the BOXMOX simulations is described in Section 2.1.1. We focus the analysis here on DMS and its major oxidation products and the effects of temperature and [$NO_x$] on these. Section 3.1 compares DMS mechanisms based around the CS2 and ST schemes used in UKCA (**Table 1**). In Section 3.2 our newly developed mechanism is compared to other DMS mechanisms from recent literature that also include HPMTF formation.

### 3.1 Comparison of DMS mechanisms used for UKCA

#### 3.1.1 Time series analysis

The BOXMOX set up allows a steady-state to be achieved for a number of key sulfur species with the main exception being $H_2SO_4$, which builds up over time in the model as the model is run without aerosol formation and aerosol microphysics included (**Figure 2**). The DMS concentration simulated with different DMS mechanisms used in UKCA is simulated to be very similar throughout all model runs; the small variations stem from different oxidant concentrations or small differences in the rate constants used for the initiation reaction in the different mechanisms (**Figure 2a**). For instance, the ST run has higher DMS concentration because the $NO_x$ concentration is lower (as is OH) and less DMS is oxidised.

The $SO_2$ concentration is increased and MSA is significantly decreased in the updated CS2 runs (CS2-HPMTF and CS2-UPD-DMS) compared to CS2 (**Figure 2b,c**). Comparing CS2-HPMTF and CS2-UPD-DMS, we can see that this is due to reaction 7c, which directly forms $SO_2$ and suppresses $CH_3SO_2$, consequently lowering MSA formation. The $SO_2$ concentration is lower in CS2-HPMTF compared to CS2-UPD-DMS because the addition of HPMTF produces OCS which acts as a longlived sulfur reservoir. While MSA concentration is very similar between CS2 and ST, $SO_2$ concentration is not. This is primarily explained through the difference in the treatment of MSA and $SO_2$ production in CS2 and ST. MSA is not treated as a reactive species in CS2 and ST (in so much as there are no further reactions of MSA after its production). In ST and ST~CS2, 100% of DMS yields $SO_2$, regardless of the amount of MSA production. However, as more MSA is produced in CS2 the $SO_2$ yield is lowered. In spite of higher $SO_2$ concentrations in the ST DMS schemes, this trend does not translate to $H_2SO_4$ concentration (**Figure 2d**). $SO_2$ is a relatively long-lived species and can therefore be lost through the mixing processes with the background air in the BOXMOX setup. In CS2, $CH_3SO_3$ decomposition provides a direct pathway to $H_2SO_4$ production. In the updated CS2 schemes (CS2-UPD-DMS and CS2-HPMTF) $SO_3$ production with instantaneous transformation to $H_2SO_4$ is included. The slower rate constant in CS2 for the decomposition of $CH_3SO_3$ (11b_old) is compensated by a higher production of $CH_3SO_3$.



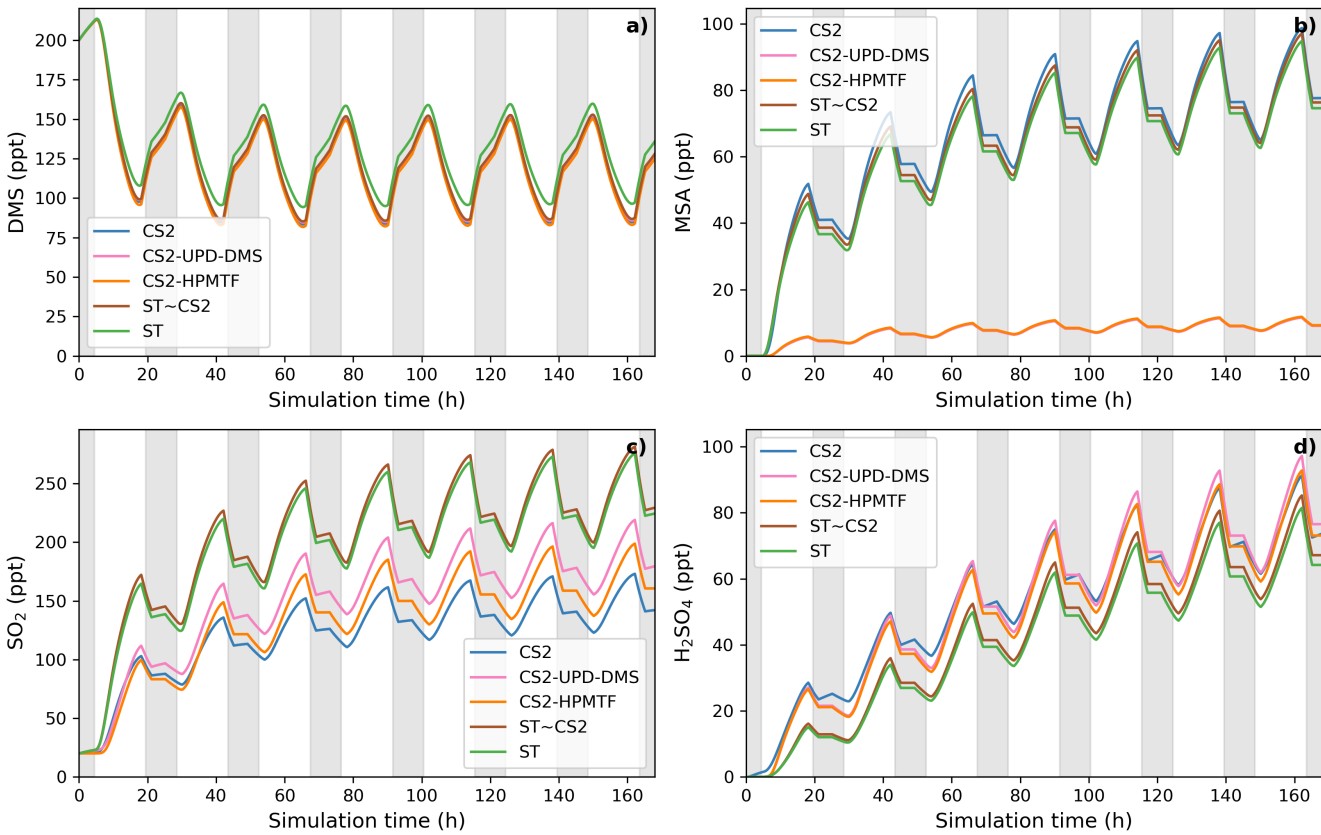

**Figure 2:** BOXMOX-simulated gas-phase concentrations as a function of time for a selection of species simulated with the different DMS gas-phase oxidation schemes used in UKCA configurations (oxidation by OH and $NO_3$). Grey areas denote nighttime, when no photolysis reactions are taking place. Average $NO_x$ concentration is approximately 10 ppt, with an average temperature of 293 K (range: 289 - 297 K).

### 3.1.2 Sensitivity of UKCA DMS schemes to temperature

As described in Section 2.1.1, a series of BOXMOX experiments were performed perturbing the temperature profile in the model (**Figure 3**).

As temperature increases in the box model, the steady state DMS concentration increases in all simulations. This is mainly because the DMS oxidation by OH addition is negatively temperature dependent. For most models, DMS concentration increases by 85-93 ppt throughout the temperature range from 260 K to 310 K, except the ST run where at temperatures over 290 K, a stronger increase of DMS concentration is found, with a total increase of 106 ppt. This could be due to different





oxidant concentrations in the model runs using the ST mechanism and independent of the DMS scheme since this stronger
increase is not found with CS2-ST.
Although the kinetics, and therefore temperature dependence, of DMS loss is comparable across the different schemes, the
dependence of MSA and $SO_2$ on temperature differ significantly.

Most MSA is formed from the OH-addition channel, which is favoured at low temperatures (Barnes et al., 2006). Therefore,
the MSA concentration is higher at lower temperatures across all the UKCA DMS schemes considered (**Figure 3b**). In the ST
schemes (ST and ST~CS2), MSA decreases by around 88% (-189 ppt and -197 ppt) throughout the temperature range
considered, while in all the CS2 schemes MSA is shown to be much more sensitive to temperature, decreasing by >96% (CS2:
–300 ppt, CS2-UPD-DMS: -222 ppt, CS2-HPMTF: -222 ppt ). In particular the CS2 family of mechanisms shows pronounced
temperature sensitivity between 270 to 290 K. We attribute this to differences in the rate constant of DMS oxidation through
the OH-addition channel (see **Table 2** and **S1.3.1**). The average MSA concentration for the UKCA schemes diverges most in
the temperature range between 270 - 300 K.

The difference in $SO_2$ concentrations between the CS2 schemes and ST schemes are greatest at lower temperature (**Figure**
**3c**), with the ST and CS2-ST schemes simulating ~ 5 times (+200 ppt) the $SO_2$ that is simulated in the other schemes based
around CS2. In the ST schemes $SO_2$ concentration either stays at a similar level across the whole temperature range (ST: +3%)
or slightly decreases (ST~CS2: -9%). Conversely, the CS2 family of schemes show a positive temperature dependence (i.e.,
$+\frac{d[X]}{dT}$), across the temperature range, especially in the range of relevant atmospheric temperatures from 270 to 290 K. $SO_2$
increases by 298% in CS2, 84% in CS2-UPD-DMS and 79% CS2-HPMTF. In the CS2 schemes, more DMS reacts through
the addition pathway which favours the production of MSA, instead of $SO_2$ therefore reducing the $SO_2$ concentration. In ST,
the addition pathway still leads to 100% $SO_2$ formation, making the average $SO_2$ concentration less dependent on temperature.
Experimental findings (Arsene et al., 1999) and field measurements (Sciare et al., 2001) both show a positive temperature
dependence of $SO_2$ concentration. This trend is only reproduced by the DMS schemes based on the CS2 mechanistic features
(i.e. not the very simple mechanism used in ST), indicating that the ST DMS chemistry is likely insufficient to explain
laboratory and field observations, particularly in cold environments and under climate change.

In these box model experiments only gas phase losses and mixing of species with background air are considered. Under the
conditions of our simulations, we find that the MTMP isomerization pathway mainly yields $SO_2$, as does the rest of the
abstraction pathway. Therefore, the addition of the isomerization branch does not have a significant impact on the temperature
dependence of $SO_2$ concentration (comparing CS2-UPD-DMS and CS2-HPMTF), even though the isomerization step itself is
greatly temperature dependent.





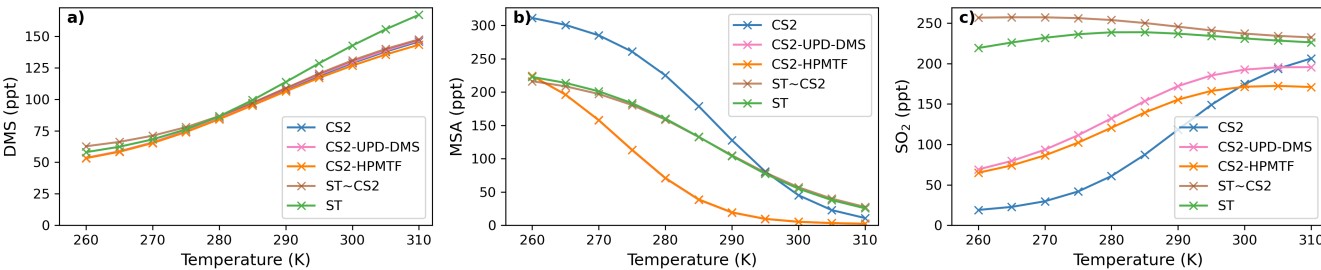

**Figure 3:** Temperature dependence of average a) DMS, b) MSA, and c) $SO_2$ concentration after a steady-state is reached in the box model simulations using the DMS schemes for UKCA.

## 3.2 Comparison with DMS schemes that include HPMTF from the recent literature

Here, four recently published DMS schemes that also include the isomerization pathway and formation of HPMTF are compared with our new mechanism, CS2-HPMTF (*CS-H*, 36 reactions in DMS scheme), as follows. To make the studies comparable, only DMS oxidation by $NO_3$ and OH and gas-phase reactions are considered. The implementation of these chemical schemes in BOXMOX can be found in the *Supporting Information S1.3*.

- *Fung et al. (2022) (FG):* This scheme includes 32 reactions for the DMS oxidation chemistry. The H-abstraction pathway is based on the MCM, while the rate constants in the OH-addition pathway mostly stem from Burkholder et al. (2015) or a scaled up version of those. The rate constant of MTMP isomerization to HPMTF is based on Veres et al. (2020).
- *Wollesen de Jonge et al. (2021) (WJ):* This scheme is the most complex and consists of 98 reactions, including reactions from the MCM and from Hoffmann et al. (2016). The isomerization branch mostly uses the rate constants by Wu et al. (2015), except the first isomerization rate constant, which is a combination of Veres et al. (2020) and Berndt et al. (2019).
- *Khan et al. (2021) (KH):* This scheme is based on Khan et al. (2016), which is equivalent to the DMS chemistry in CS2 (CRI v2 R5). The mechanism was modified to include the isomerization pathway and photolysis loss and temperature dependent OH oxidation of HPMTF by the authors. In total, the DMS chemistry consists of 38 reactions, 5 of which are photolysis reactions.
- *Novak et al. (2021) (NV):* This is a simplified scheme that aims to only include the intermediates necessary for HPMTF formation and consists of only 10 reactions. DMS therefore either directly yields MSA (without DMSO formation) or first forms MTMP, which isomerizes to form HPMTF or is oxidised to $SO_2$.





·21   Using this ensemble of gas-phase DMS oxidation schemes in BOXMOX simulations leads to significant differences in the

·22   concentrations of important oxidation intermediates and products, even though DMS concentration is similar across all models

·23   (**Figure 4**).

·24   **3.2.1 Time series analysis of different DMS-HPMTF schemes**

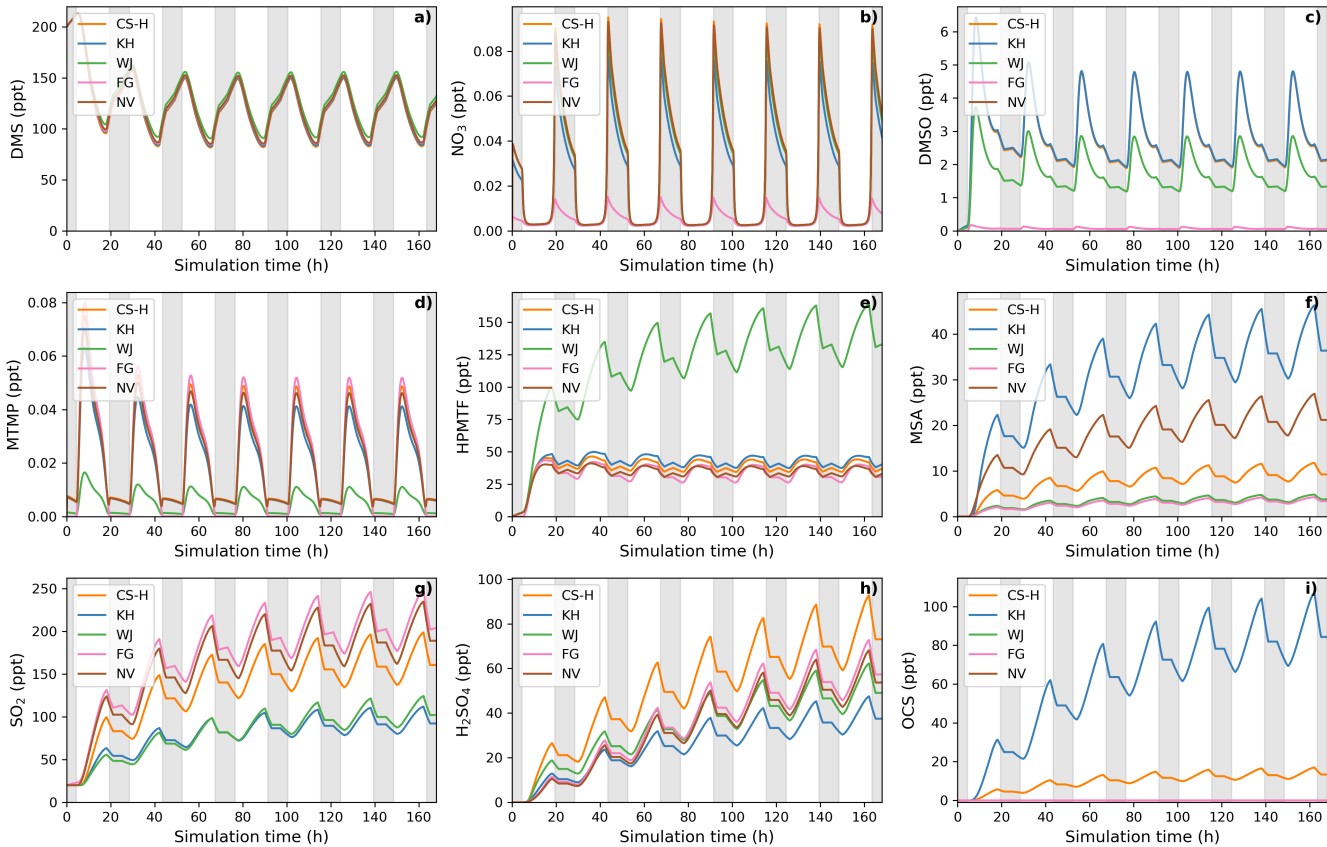

·25

·26   **Figure 4:** Gas-phase concentrations as a function of time for different DMS gas-phase oxidation schemes (oxidation by OH

·27   and $NO_3$). Average $NO_x$ concentration is approximately 10 ppt, with an average temperature of 293 K (range: 289 - 297 K).

·28   Grey areas denote nighttime when no photolysis reactions are taking place.

·29

·30

·31   The depletion of DMS due to OH and $NO_3$ oxidation is similar across most models (**Figure 4a**) since the major oxidants are

·32   relatively constrained by the box model experiment set up (see Section 2.1.1) and they mostly rely on IUPAC or JPL

·33   recommended values (Atkinson et al., 2004; Burkholder et al., 2015). One exception is the $NO_3$ oxidation in the FG scheme,

·34   which is a factor of approximately 6 higher than the JPL rate constant. On the one hand, this does not affect DMS concentration,

·35   since OH oxidation of DMS plays a greater role, on the other hand, the concentration of $NO_3$ in the FG scheme's simulation





run is controlled by the greater $NO_3$ oxidation rate (**Figure 4b**). WJ includes the intermediate $CH_3S(OH)CH_3$ and its
decomposition back to DMS (based on Hoffmann et al., (2016)), which in their experiments improved the fit between their
measured and modelled DMS concentration. Here, this does not have any significant impact on DMS concentration, compared
to all the other schemes.

Significant differences between the models can be found for the DMSO concentration (**Figure 4c**). KH and CS-H have the
highest DMSO concentration since all DMS that is oxidised through the OH-addition pathway yields DMSO. This is not the
case for WJ, where $CH_3SOH$ and to a small part DMSO2 are also possible products. In the FG simulation, DMSO concentration
is close to zero, which is due to a much faster loss of DMSO;  a rate constant a factor of 15 faster than experimental
measurements by Urbanski et al. (1998). NV does not include DMSO as an intermediate. Since the lifetime of DMSO was
found to be several hours (Urbanski et al., 1998; Ye et al. 2021), deposition of DMSO could act as a significant sink of
atmospheric sulfur (as found by Chen et al. (2018)). Fast oxidation of DMSO in FG, or omitting the  species in NV, might
therefore lead to an over-estimation of other DMS oxidation products in those schemes.

Regarding the intermediate MTMP, WJ shows the greatest deviation from the ensemble (**Figure 4d**). The MTMP concentration
never exceeds 0.02 ppt in WJ, while the other mechanisms simulate concentrations over three times higher. WJ employs a
faster isomerization rate constant of MTMP to HPMTF. They scale the A-factor by 5 to get a rate constant that is a combination
of the theoretical calculations by Veres et al. (2020) and the experimental findings by Berndt et al. (2019). Additionally, they
include more oxidation reactions of MTMP (such as oxidation by $NO_3$) but since the isomerization to HPMTF already
outcompetes most oxidation reactions anyway (>97%), we found them to play a negligible role (<0.1%). In the FG scheme,
DMS + $NO_3$ leads to immediate $SO_2$ formation, without prior MTMP formation. Therefore, no MTMP is produced during the
nighttime, when $NO_3$ oxidation becomes relevant. Under conditions with low $NO_x$ (around 10 ppt in this experiment) this does
not have significant impacts but at higher $NO_x$ concentrations this leads to a major deviation from the other simulations (**Figure**
**5a**, 100 ppt $NO_x$). At night, CS-H, KH, and NV reach MTMP concentrations of 0.07 ppt, allowing nighttime HPMTF
formation, while FG stays zero.

All model simulations, except WJ, are very similar in HPMTF concentration (**Figure 4e**). The fast isomerization rate constant
in WJ is one of the reasons HPMTF concentration is on average more than 3 times higher than the other model simulations.
The other reason is a much slower oxidation of HPMTF by OH. While most models use a value of (or close to) $1.11 \times 10^{-11}$
$cm^3$ molecule$^{-1}$ s$^{-1}$, recommended by Vermeuel et al. (2020), WJ use the much slower rate constant calculated by Wu et al.
(2015), $1.4 \times 10^{-12}$ $cm^3$ molecule$^{-1}$ s$^{-1}$. This rate constant is also used in the KH scheme but it additionally includes HPMTF
depletion by photolysis which ultimately leads to the similar HPMTF concentration as in CS-H, FG, and NV. The addition of
the photolysis reactions in KH does not affect the diel profile of HPMTF, even though those account for 81% of chemical loss
of HPMTF in their scheme. It is therefore unlikely that the observed diel profile of HPMTF by Vermeuel et al. (2020) and





Khan et al. (2021) can be explained solely by considering loss of HPMTF to aldehyde and hydroperoxide photolysis. Reducing
HPMTF formation to one isomerization reaction without any side reactions as is done in this work and NV, does also not affect
the diel profile of HPMTF significantly.
The effect of higher $NO_x$ conditions on the diel profile of HPMTF varies significantly between the different schemes (10 ppt
$NO_x$ in **Figure 4** vs. 100 ppt $NO_x$ in **Figure 5**). Higher $NO_x$ concentration leads to more DMS oxidation by $NO_3$ at night and
the subsequent increase in MTMP concentration and therefore HPMTF concentration during the night hours in the CS-H, WJ,
KH, and NV simulations. At low $NO_x$, HPMTF concentration stayed more or less stable throughout the nighttime and increased
in the morning, reaching a plateau in the afternoon, and dropping in the evening (**Figure 4e**). Under higher NOx conditions,
HPMTF increases in these mechanisms throughout the night and decreases throughout the day when it is oxidised by OH
(**Figure 5b**). In the WJ simulation, the diel profile has more plateaus and small deviances but the overall trend still fits the
described pattern. This is not true for FG, where DMS oxidation by $NO_3$ leads directly to $SO_2$ formation.

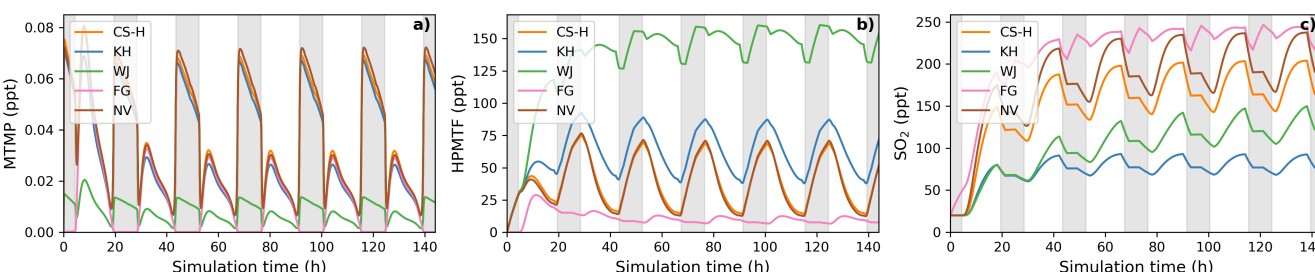


**Figure 5:** BOXMOX simulations where the average $NO_x$ concentration is approximately 100 ppt (a factor 10 greater than for
the results presented in **Figure 4**). **(a)** MTMP, **(b)** HPMTF, and **(c)** $SO_2$ concentration as a function of time for different DMS
gas-phase oxidation schemes (oxidation by OH and $NO_3$). Average temperature of 293 K (range: 289 - 297 K). Grey areas
denote nighttime when no photolysis reactions are taking place.

While the diel profile of MSA looks similar for all simulations, the average concentrations do not (**Figure 4f**). The highest
average steady-state MSA concentration is reached in the KH simulation, which is a factor of 10 higher than the lowest average
concentration in the FG simulation. In our experimental setup, most of the simulations we performed with the different
mechanisms do not include any (significant) gas-phase chemical loss pathway for MSA; MSA is only lost through mixing and
transport out of the "box". Therefore, the concentration of MSA is a direct reflection of MSA production in the respective
simulations.

KH simulates the highest production of MSA (similar to CS2), where MSA is formed through the addition (MSIA + OH →
0.05 MSA + 0.95 $CH_3SO_2$, reaction 9b,c) and the abstraction channel ($CH_3SO + O_3$ → $CH_3SO_2$, reaction 7c_old) of DMS
oxidation, with $CH_3SO_2$ partly being oxidised to $CH_3SO_3$ and then to MSA (reactions 10b,c, 11a). The decomposition of



CH$_3$SO$_3$ to H$_2$SO$_4$ in KH is slower than in other mechanisms, increasing the branching ratio for MSA formation in their
mechanism. In NV, the simulation with the second highest average MSA concentration, the only source of MSA is the direct
production of MSA through OH oxidation through the addition channel, where 25% of DMS forms MSA. In both, CS-H and
WJ, the abstraction pathway mostly produces SO$_2$ and only contributes negligible amounts to CH$_3$SO$_2$ formation, hence MSA.
Similar to KH, the oxidation of DMS through the addition pathway in CS-H and WJ yields CH$_3$SO$_2$ of which a part forms
MSA. However, not all of the CH$_3$SO$_2$ results in MSA, some of it also decomposes to SO$_2$ or yields SO$_3$. This explains the
lower concentration of MSA in CS-H and WJ compared with NV. The reason why CS-H has a higher MSA concentration than
WJ is because of the inclusion of reaction 9b (**Table 2**), which yields MSA directly and is not part of the WJ scheme.
The lowest MSA concentration is found in FG and WJ, where 60% of the OH-addition pathway directly produces SO$_2$. Out of
the 40% of DMS that forms DMSO in this pathway, only a fraction yields MSA.

To harmonise the results and aid interpretability, the same rates (based on CS2) are used for the loss processes of SO$_2$ in all
the mechanisms considered here, therefore the concentration of SO$_2$ can be used as a proxy for SO$_2$ production, just as for
MSA. The highest SO$_2$ concentration can be seen in schemes that have the smallest number of intermediates or the most direct
pathways from DMS to SO$_2$, in NV and FG (**Figure 4g**). Fewer intermediates result in less opportunities for the formation of
side products or less long-lived species that can be lost through transport or deposition. For instance, in WJ HPMTF is lost
through mixing with the background before it can form SO$_2$. Likewise, KH has a higher ratio of MSA and OCS production,
which lowers the SO$_2$ yield. The diel profile of SO$_2$ concentration is in most simulations not affected by higher NO$_x$
concentrations, with the general trend being an increase of SO$_2$ concentration during the day and a decrease at night (**Figure**
**5c**). The only exception is the FG simulation, where we see a clear increase through part of the night, due to the reaction DMS
+ NO$_3$ → SO$_2$ .
The H$_2$SO$_4$ concentration is influenced by SO$_2$ production and CH$_3$SO$_3$ production and the rate of decomposition of SO$_3$ to
H$_2$SO$_4$. CS-H has the highest average H$_2$SO$_4$ concentration and KH the lowest; all other models are very similar to each other
(**Figure 4h**). In general, higher SO$_2$ concentration leads to more H$_2$SO$_4$, since SO$_2$ is first oxidised to SO$_3$ and then to H$_2$SO$_4$
with the same rates across all schemes. However, all models except NV include an additional pathway of H$_2$SO$_4$ formation: in
KH and FG, H$_2$SO$_4$ is directly formed from CH$_3$SO$_3$, while in CS-H and WJ CH$_3$SO$_3$ decomposes to SO$_3$ first, which then
instantly reacts to H$_2$SO$_4$. In KH, the rate constant for the decomposition of CH$_3$SO$_3$ at 295 K is a factor of 15 slower than in
the other models. Since the SO$_2$ concentration is also relatively low, it explains why KH has the lowest H$_2$SO$_4$ concentration
of all schemes when reaching steady-state. CS-H results in a higher H$_2$SO$_4$ concentration than FG or NV even though those
models have a higher SO$_2$ concentration. The reason is a higher production of CH$_3$SO$_3$ that is then decomposed to SO$_3$ and
H$_2$SO$_4$.





Similar to the other products of the DMS scheme, the concentration of OCS is a reflection of its production. OCS is only
produced from oxidation of HPMTF by OH and, in the KH scheme, through photolysis of HPMTF. In KH, 60% of HPMTF
forms OCS, resulting in the highest OCS concentration (**Figure 4i**). This stems mainly from the large contribution of the
photolysis reactions. Potentially, the rate constant of OH oxidation of HPMTF in KH is too low and therefore OCS might be
overestimated. In CS-H, 10% of HPMTF is oxidised to OCS, resulting in an OCS concentration that is on average 5.5 times
lower than KH. FG and WJ both use the theoretically determined branching ratio by Wu et al. (2020), which results in only
0.007% of HPMTF being oxidised to OCS at 295 K. NV does not include this pathway. Very recent evidence suggests that
there is a small (2%) but prompt source of OCS following the formation (and decomposition) of HPMTF as well as a significant
OCS yield (13%) from the HPMTF + OH reaction (Jernigan et al., 2022). These new data were not assessed (or included) in
this work but we estimate that inclusion of these mechanistic pathways would result in OCS yields between CS-H and the
other mechanisms i.e. consistently lower than that simulated by KH.

### 3.2.2 Temperature dependence of different DMS-HPMTF schemes

**Figure 6** shows that even though the temperature dependence of average DMS concentration is similar across all schemes, the
temperature dependence of average $SO_2$ and MSA concentration differs from scheme to scheme significantly. Most of the
general trends were found to be similar and in line with the trends observed for the UKCA schemes and have been explained
there (Section 3.1.2, **Figure 3**).
While WJ has the highest absolute change in HPMTF concentration throughout the temperature range (+131 ppt, +380%;
**Figure 6b**), CS-H, KH, and NV show higher relative change (+43-48 pp, +763-892%). Since FG is missing the DMS oxidation
by $NO_3$ as a potential pathway to HPMTF (via MTMP), HPMTF in FG is least affected by temperature (+34 ppt, +256%).

MSA is even more affected by temperature than HPMTF (**Figure 6c**). Its concentration shows a strong negative temperature
dependence in all simulations (**Figure 6c**). The magnitude of MSA-temperature dependence differs from scheme to scheme.
The smallest changes can be observed in NV (-47 ppt from 260 - 310 K), where only 25% of DMS that is oxidised through
the OH-addition pathway forms MSA. Similarly in FG (-67 ppt from 260 - 310 K), where only 40% of the OH-addition
pathway forms DMSO and then potentially MSA. The largest temperature dependence can be found in the KH simulation,
with a change of MSA concentration of -282 ppt from 260 K to 310 K, which is very similar to CS2 (**Figure 3c**).

In almost all schemes, $SO_2$ concentration increases with temperature (**Figure 6d**). The greatest positive change happens
between the atmospheric relevant temperatures 270 and 290 K. KH and CS-H show the greatest increase in this temperature
range with +53 ppt (+160%) and +69 ppt (+80%), respectively (WJ: +34 ppt (51%)). Starting at 295 K, $SO_2$ concentration



plateaus with further increasing temperature and even declines slightly in some simulations (Figure 6d). NV and FG are the only models which show a decrease in $SO_2$ throughout the entire temperature range of 260 - 310 K (NV: -24 ppt, -11%, FG: -22 ppt, -10%), similar to ST~CS2 in **Figure 2d**. This could be due to previously mentioned simplifications in the DMS additional channel, where DMSO is either completely omitted or rapidly oxidised further.

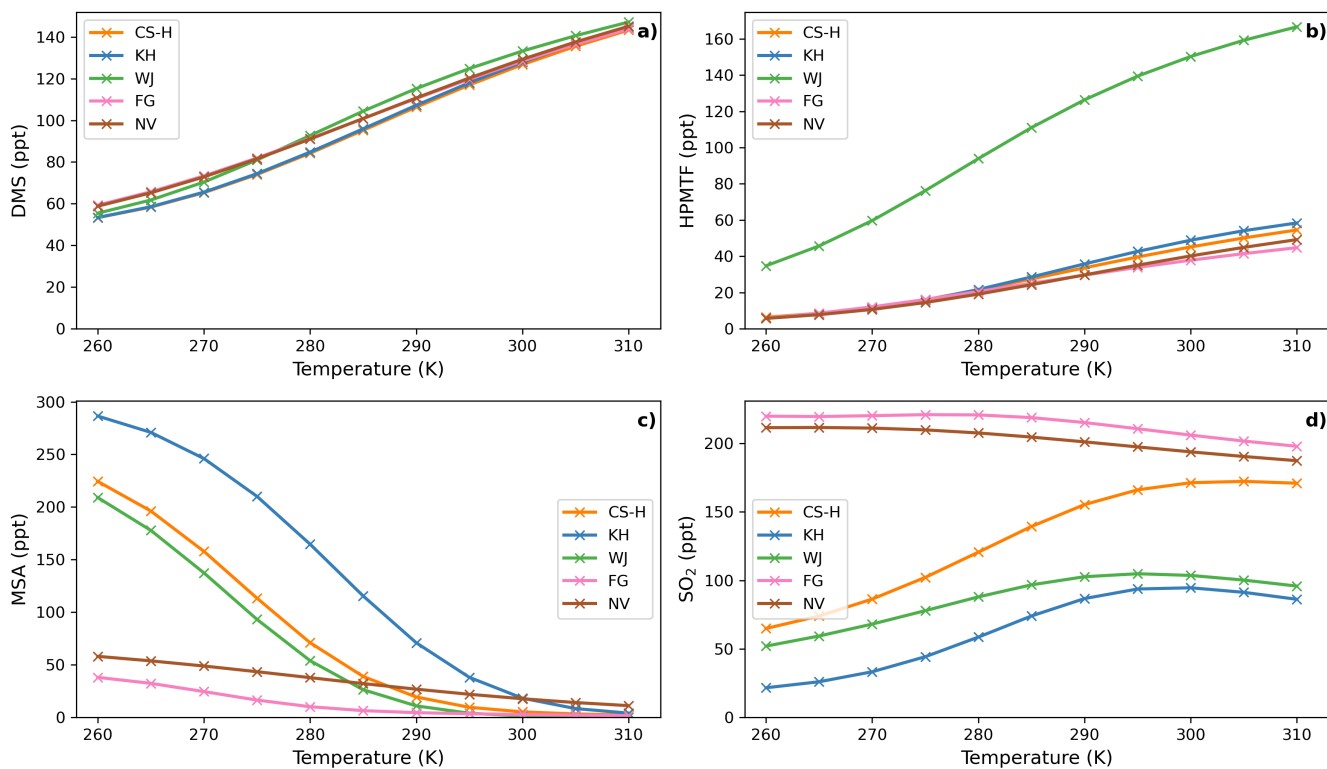

**Figure 6:** Temperature dependence of average **(a)** DMS, **(b)** HPMTF, **(c)** MSA, and **(d)** $SO_2$ concentration in different DMS oxidation schemes after a steady-state is reached in the box model simulation. Average NOx is approximately 10 ppt.

These results demonstrate limited consensus on gas-phase DMS oxidation, similar to the earlier work of Karl et al., 2007. Importantly in the context of the role of DMS in chemistry-aerosol-climate feedbacks, we have further shown that this uncertainty across mechanisms is amplified when assessing temperature sensitivity of the products of DMS oxidation. Small uncertainties in the rate of reactions or the omission of intermediates can have significant effects on the resulting product concentrations. At present there is a need for more laboratory data and more focused sensitivity studies to isolate the major sources of uncertainty that are common across DMS oxidation mechanisms and constrain them. Strikingly we see that the ST and CS2 mechanistic variants used for UKCA studies span the wide range of SO2-Temperature and MSA-Temperature





'80  sensitivities as the recently reported updated DMS mechanisms. We now move on to discuss our work implementing the CS2-

'81  H mechanism into our global chemistry-climate model.

## 4 Results from 3D model simulations using UKCA

'83  Here we present our results from the incorporation of the new CS2-H DMS mechanism described above in the 3D UKCA

'84  chemistry climate model. As described in Section 2.1, we performed a series of 12 month nudged simulations with UKCA for

'85  the year 2018 using 6 model simulations, with different mechanistic variants (**Table 1**). As a reminder, we use the CS2

'86  simulation (Archer-Nicholls et al., 2021) as the "base" simulation, to which mechanistic improvements are made.

### 4.1 Distribution of DMS

'88  The annual mean global DMS burden was found to be between 63-66 Gg S in all model simulations. DMS concentration

'89  follows a seasonal modulation with maximums in the warmer months, which coincide with phytoplankton blooms (**Fig. 7a**).

'90  **Figure 7b** and **7c** show the annual mean vertical profiles in the central North Atlantic region and the Southern Ocean (see

'91  figure caption for bounding areas). These regions are focused on owing to the differences shown in the mixing ratios of key

'92  species and the importance of these two regions to global climate (e.g., Sutton et al., 2018; Caldeira and Duffy 2000). In the

'93  Southern Ocean, DMS mixing ratios vary between 100 and >300 ppt. On the other hand, in the North Atlantic region analysed,

'94  DMS concentrations rarely reach over 50 ppt. Here, <1 ppt DMS is found above the boundary layer (above 1000 m), while in

'95  the Southern Ocean DMS decreases more slowly up-to the tropopause (~8000 m). These differences in DMS distribution are

'96  a complex function of the local heterogeneity of the DMS source from the ocean and differences in the lifetime of DMS due

'97  to different simulated cloud and oxidising environments (with the North Atlantic generally being a region of greater oxidising

'98  capacity than the Southern Ocean (Archer-Nicholls et al., 2021; Griffiths et al., 2021))

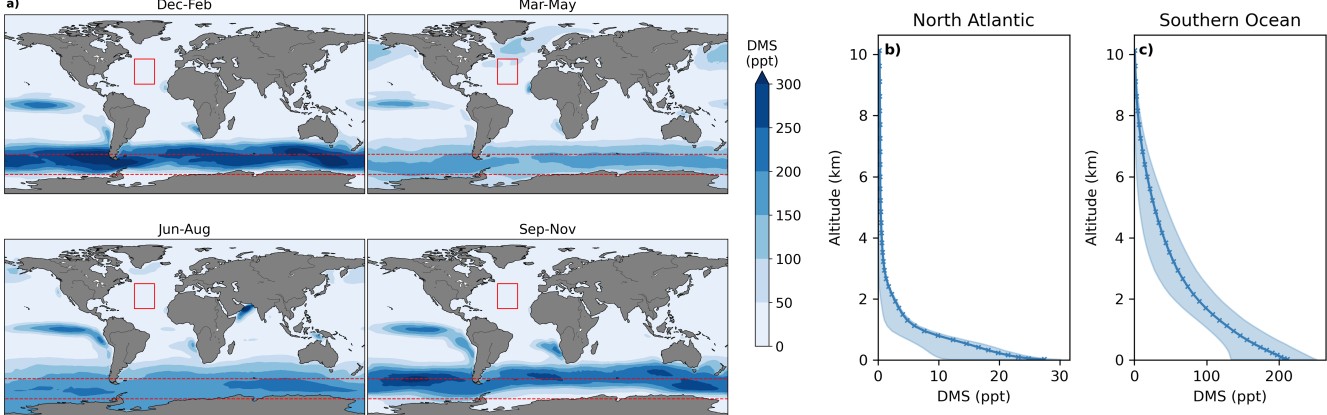

'01  **Figure 7**: **a)** Global distribution of DMS mixing ratios in the lower troposphere (< 2 km) over the oceans in CS2. Annual mean





vertical distribution of DMS in **b)** the Central North Atlantic (30-50°E, 20-45°N, denoted with the red rectangle in panel **a)** and in **c)** the Southern Ocean (50-70°S, denoted with the red dashed rectangle in panel **a**). The envelopes represent the interquartile range of the model simulation results. Note the order of magnitude difference in the DMS concentrations between the North Atlantic and Southern Ocean.

### 4.1.1 Comparison of DMS with observations

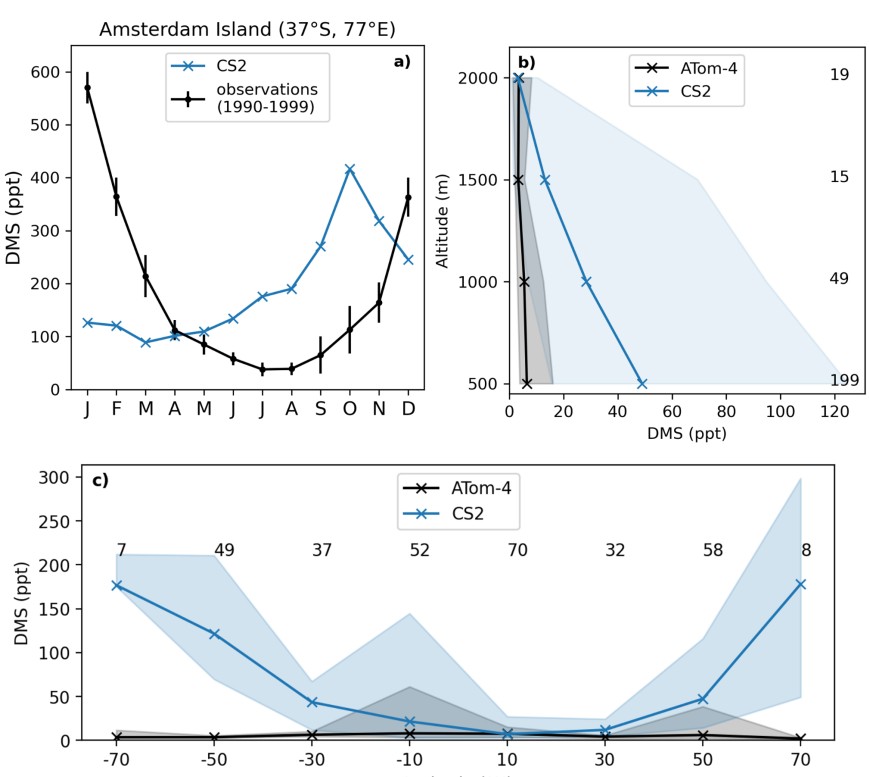

**Figure 8: (a)** Comparison of DMS surface concentration on Amsterdam Island (37°S, 77°E) in the southern Indian Ocean. The observational data (Sciare et al., 2000) represents the monthly mean concentrations and their standard deviations for the years 1990-1999. **(b)** Vertically binned (500 m) and **(c)** latitudinally binned (20°) median DMS mixing ratio along the ATom-4 flight path. The envelopes represent the interquartile range of the measurements and the respective model results while the numbers on the side/on top give the number of measurements in the respective bin. In b) and c) the model data are sampled along the ATom flight track using hourly mean model data.





DMS was not significantly affected by the different DMS mechanisms in the simulations with UKCA and so we focus on the results from the CS2 simulation. **Figure 8** compares observed DMS from ground based measurements on the Amsterdam Island and *in situ* measurements from the ATom-4 flights with the simulated DMS in CS2. On Amsterdam Island, a clear seasonality was observed for the monthly mean DMS concentration, with a peak during the austral summer (570 ppt) and a minimum during the austral winter (38 ppt). The simulated DMS (89 - 416 ppt) falls within that range but fails to capture the observed seasonal trends (**Figure 8a**). We suggest that the disagreement between the observed and modelled atmospheric DMS mixing ratios is driven by the DMS emissions dataset we have applied in this study. Bock et al. (2021) reviewed CMIP6 DMS emissions and found that the emissions used in our study (from the UKESM-1 model) tend to result in less spatial heterogeneity than observational based climatologies (e.g., Lana et al. (2011)). During the tuning of the UEKSM-1 model (Sellar et al., 2019) the DMS emission scheme was modified to have a minimum DMS ocean concentration of 1nM imposed, the effect of which seems to be to generally overestimate the DMS emissions over the low productivity regions. Overall, we found the atmospheric DMS concentration in the model (which hitherto has not been evaluated before) to be significantly higher compared with the airborne observations from ATom-4 (**Figure 8b** and **c**). At the altitudes shown, the model predicts DMS approximately 5 times higher than the measurements. A comparison along the latitudinal axis (**Figure 8c**) reveals that DMS is significantly overestimated at high latitudes (however, it should be taken into account that only few measurements exist for latitudes above 60° from ATom4).

It is difficult to evaluate atmospheric DMS globally as there are limited observations that can be used for evaluating global models. For instance, Amsterdam Island being one of only a handful of long-term observational sites, no remote sensing based data and with most atmospheric observations made on ships that are focusing on plumes of DMS. None the less, our CS2 base run (and all subsequent UKCA runs) suffer from a high bias in simulated atmospheric DMS, driven by the use of the emissions dataset we used. We opted to use the default UKESM DMS emissions as our focus in this study is the oxidation mechanism. However, we suggest that future work assess the impacts of both DMS emissions and chemistry using some of the more recent DMS emissions datasets (Gali et al., 2018; Hulswar et al., 2022). Bearing the caveats of DMS in mind, we now look at the intermediates and products of DMS oxidation.

### 4.1.2 Oxidation of DMS

We calculate a global average tropospheric lifetime of 1.5 days for DMS. **Figure 9** shows the global distribution of the different DMS oxidation pathways in the base run (these results are not affected by the different DMS mechanism variants we use as these reactions were not updated and there is only a weak feedback of DMS oxidation products on DMS oxidation itself). 75% of DMS is oxidised by OH (41% via the OH-addition channel and 34% via the H-abstraction channel) and 25% by $NO_3$. Oxidation by $NO_3$ is dominant in the Northern Hemisphere, especially close to the coast and over ship routes. In the Southern



Hemisphere, where DMS emissions are highest, the contribution is less than 20%. The addition pathway of OH oxidation is

favoured at lower temperatures, explaining the trend of higher DMSO formation at high latitudes.

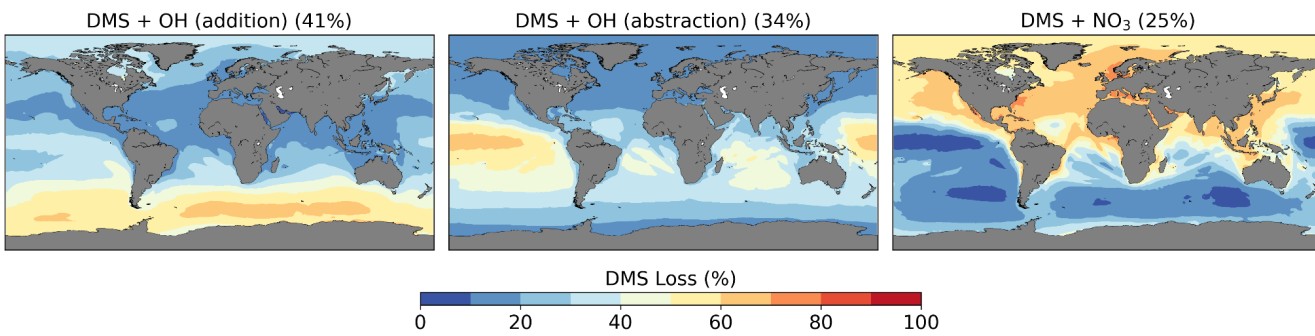

**Figure 9:** Spatial distribution of mean percentage of DMS oxidation via DMS + OH (addition), DMS + OH (abstraction), and DMS + NO$_3$ in the CS2 base run. The percentage in brackets denotes the contribution of this channel to the global chemical loss of DMS. Only values above the ocean are shown.

## 4.2 DMS Oxidation products

59% of DMS forms MTMP, the first intermediate of the abstraction pathway. In CS2, MTMP is oxidised by NO (51%) or

reacts with itself (49%) to form CH$_3$S (**Figure 10a**) which is further oxidised to SO$_2$, H$_2$SO$_4$, and MSA. With the updates

implemented in CS2-HPMTF, 86% of MTMP isomerizes to HPMTF, while 8% is oxidised by HO$_2$, and only 6% by NO

(**Figure 10b**). The self-reaction becomes negligible with the additional loss processes of MTMP, significantly lowering MTMP

concentrations. The global tropospheric lifetime of MTMP is reduced from 26 min to less than one minute.

667



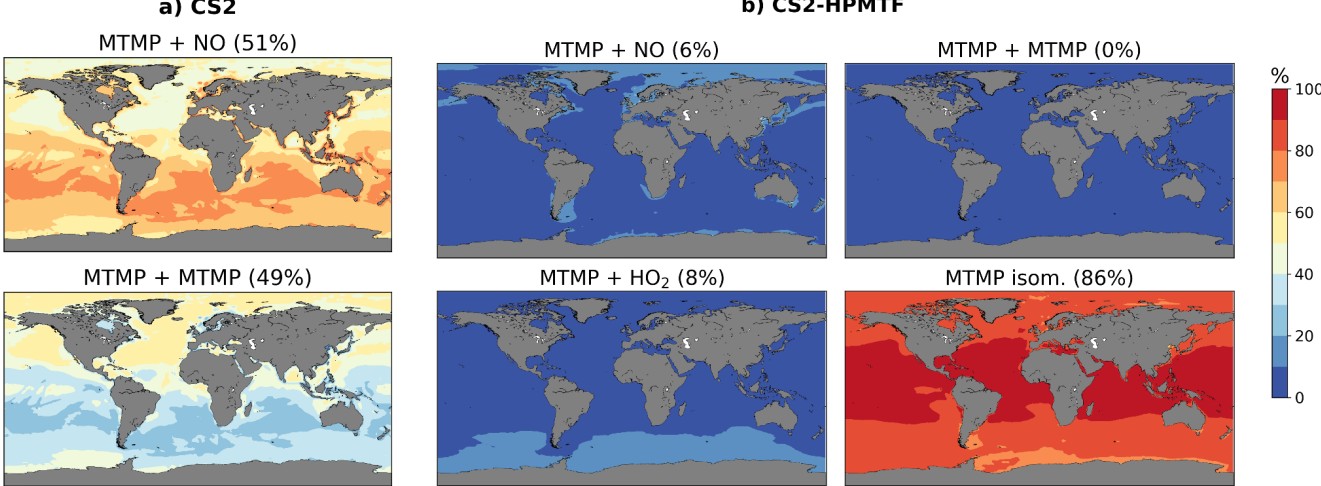

**Figure 10:** Spatial distribution of annual mean percentage of MTM depletion (< 2 km) via MTMP + NO, its self-reaction, MTMP + HO$_2$, and isomerization to HPMTF in **a)** CS2 and **b)** CS2-HPMTF. The percentage in brackets denotes the contribution of this channel to the global chemical loss of MTMP. Only values above the ocean are shown.

### 4.2.1 Modelled HPMTF

In CS2-HPMTF 51% of DMS forms HPMTF. The general patterns of the global distribution of HPMTF are similar to those of DMS in **Figure 11**, except that relatively higher concentrations of DMS are reached in the Southern Ocean. There, temperatures are lower and therefore the OH-abstraction pathway, as well as the strongly temperature-dependent isomerization reaction from MTMP to HPMTF are disfavoured. At the surface, the annual mean HPMTF concentration is similar in the North Atlantic and the Southern Ocean with approximately 20 ppt. However, in the North Atlantic, the variability throughout space and time is greater (bigger interquartile range). Further, the vertical profiles differ visibly: In the North Atlantic HPMTF concentration decreases in the boundary layer and above 2500 m HPMTF concentration is virtually zero (**Figure 11b**). In the Southern Ocean, the concentration decreases more slowly and only reaches zero at 10000 m (**Figure 11c**). The HPMTF burden in CS2-HPMTF is 24 Gg S and HPMTF has a lifetime of 26 hours



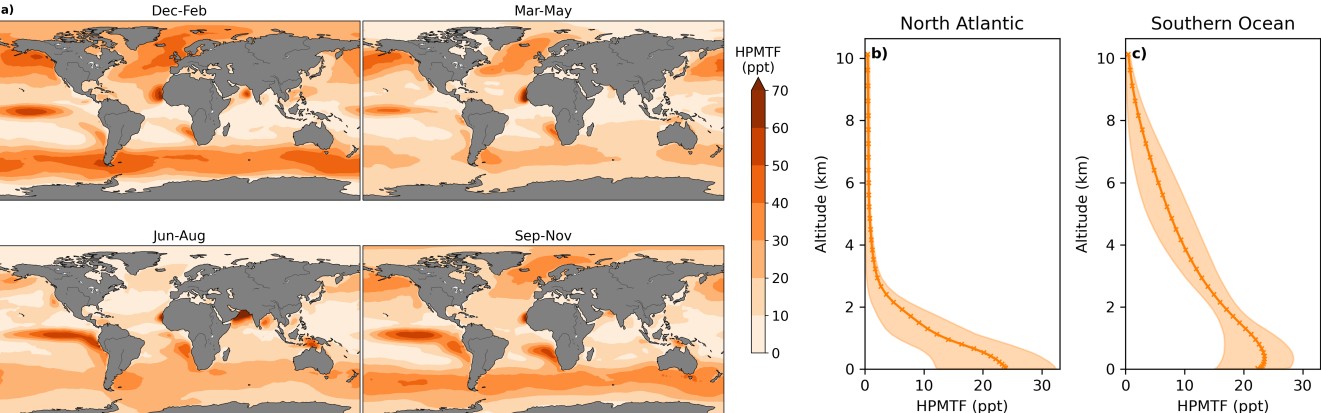

**Figure 11:** Seasonal average **a)** Global distribution of HPMTF mixing ratios in the lower troposphere (< 2 km) over the ocean in CS2-HPMTF. Annual means of the vertical distribution of HPMTF are shown in **b)** the Central North Atlantic (30-50°E, 20-45°N) and **c)** the Southern Ocean (50-70°S). The envelopes represent the interquartile range of the model data.

**Comparison of HPMTF with observations**

Since DMS in the model is likely overestimated, the same would be expected for HPMTF. **Figure 12a** shows that the implemented loss processes in CS2-HPMTF already lead to a diel profile of HPMTF that is similar to the one measured by Vermeuel et al. (2020) (where no DMS measurements were made), without the need to add aqueous loss or photolysis. While DMS at low altitudes was overestimated by a factor of 5 in the model, the maximum HPMTF is only 3.7 times higher than the highest measurement in the diel profile at Scripps Pier (**Figure 12a**). For the comparison with ATom-4 data (**Figure 12b,c**), the DMS/HPMTF is used to account for the discrepancy between DMS concentrations observed and in the model. The model generally underestimates the HPMTF/DMS ratio. For instance, up until 1000 m, the ratio in the model is half of the measured ratio. These results indicate that loss processes of HPMTF might still be too fast in the model or the oxidation of DMS too slow. The CS2 oxidants have been evaluated before (Archer-Nicholls et al., 2021) and were found to be higher in the boundary layer than in ST simulations used in CMIP6 studies but well within the spread of other models (Griffiths et al., 2021; Stevenson et al., 2020).





**Figure 12**: **a)** Comparison of the diel profile of HPMTF at the Scripps Pier at the California Coast (32°N, 117°W). The observational data (Vermeuel et al., 2020) is the mean of measurements from July 26 to August 3, 2018, while the model output is the mean from April/May 2018. **(b)** Vertically binned (500 m) and **(c)** latitudinally binned (20°) median DMS/HPMTF ratio along the ATom-4 flight path. The envelopes represent the interquartile range of the measurements and the respective model results while the numbers on the side/on top give the number of measurements in the respective bin.

**4.2.2 Modelled MSA**

MSA is an important intermediate of the OH-addition channel. It contributes to aerosol growth and might play a role in new particle formation (Chen et al., 2015; Chen and Finlayson-Pitts, 2017). MSA production is reduced from 7.9 Tg S yr$^{-1}$ in CS2 by 70% to 2.4 Tg S yr$^{-1}$ in CS2-HPMTF. In the CS2-HPMTF simulation, wet and dry deposition and gas-phase oxidation by



'14    OH to $CH_3SO_3$ have been included as loss processes for MSA, which account for 89%, 10%, and 1% of the loss of MSA;

'15    respectively. The tropospheric MSA burden is 40 Gg S in CS2-HPMTF with a lifetime of 6 days.

'16

'17    In CS2-HPMTF, MSA is greatest in the Southern Ocean (**Figure 13**), where it shows a strong seasonal pattern, similar to

'18    DMS. Mixing ratios up to 80 ppt are reached in January (Austral summer), while in July they are below 10 ppt. This is reflected

'19    in the big interquartile range of MSA in the Southern Ocean (**Figure 13c**). Since the OH addition pathway is negatively

'20    temperature dependent, MSA is primarily produced at high latitudes, inversely to HPMTF. MSA shows the greatest asymmetry

'21    in concentration between the North Atlantic and Southern Ocean out of the different species discussed here. As well as

'22    significant differences between the magnitude of MSA simulated in the North Atlantic and Southern Ocean, the vertical

'23    profiles of MSA are shown to be very different. MSA reaches a peak in concentration at around 2 km altitude in the Southern

'24    Ocean (consistent with a longer DMS lifetime and therefore greater vertical transport), whereas it peaks near the surface in the

'25    North Atlantic.

'26

'27

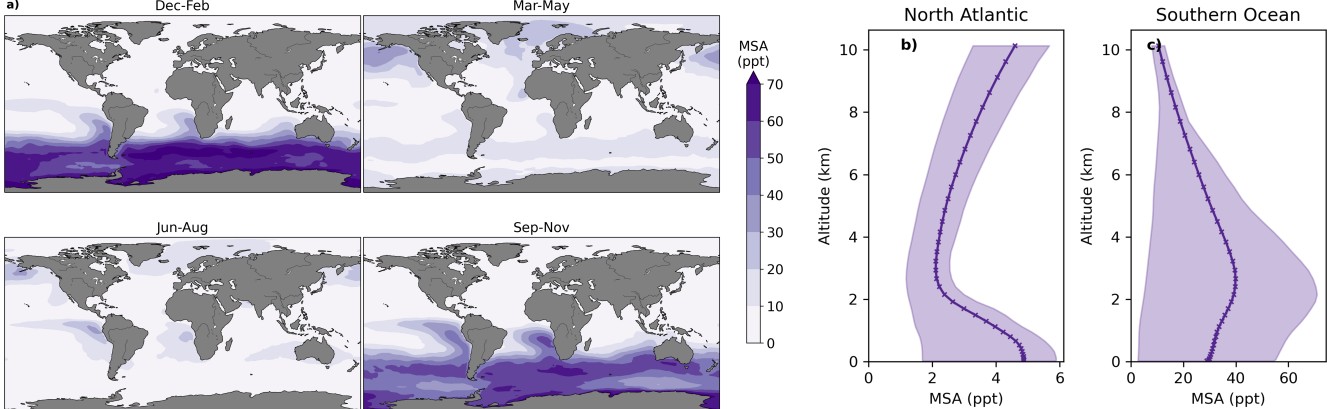

'28

'29    **Figure 13**: **a)** Global distribution of MSA mixing ratios in the lower troposphere (< 2 km) in CS2-HPMTF. Annual means of

'30    the vertical distribution of MSA are shown in the **b)** Central North Atlantic (30-50°E, 20-45°N) and **c)** Southern Ocean (50-

'31    70°S). The envelopes represent the interquartile range of the measurements. Note the order of magnitude difference in the

'32    MSA concentrations in panels **b)** and **c)**.

'33

'34    **4.2.3 Modelled SO₂ and sulfate**

'35    In CS2-HPMTF the $SO_2$ burden is increased by 5.6% compared with CS2, to 391 Gg S (**Table 4**). While this percentage seems

'36    low, a significant contribution to the $SO_2$ burden stems from anthropogenic sources and is mainly located above the land. The

'37    increase of $SO_2$ over the remote ocean, especially over the Southern Ocean, can reach up to 400% (**Figure 14**). At high





'38    latitudes, the new chemistry implemented in CS2-HPMTF also introduces a stronger seasonality to $SO_2$, whereby $SO_2$

'39    concentration is higher in respective warmer months than in CS2 (**Figure 14, 15a**). Comparison of CS2-HPMTF with ST

'40    reveals that the $SO_2$ burden is 9.2% higher in the ST run, which uses a 100% $SO_2$ yield from DMS (**Figure S7** in the SI). The

'41    global annual tropospheric sulfate burden is increased in CS2-HPMTF by 3.7% compared with CS2, to 604 Gg S. However,

'42    the sulfate burden is 5.3% higher in ST than in CS2-HPMTF (**Table 4**).

'43

'44

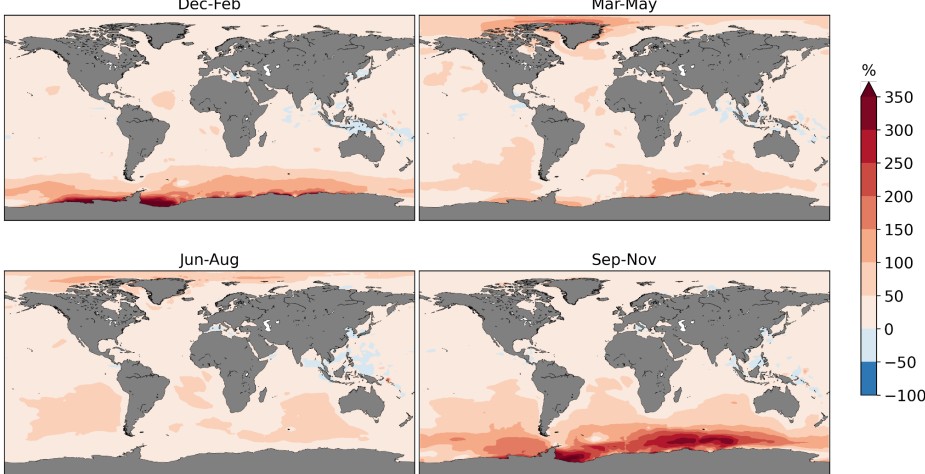

'45

'46    **Figure 14:** Relative difference in $SO_2$ mixing ratios in the lower troposphere (< 2 km) between CS2-HPMTF and the base run

'47    CS2 (CS2-HPMTF - CS2). Only values above the ocean are shown.

'48

'49    Comparing the three schemes, ST, CS2 and CS2-HPMTF, ST generally has the highest concentrations in $SO_2$ or sulfate and

'50    CS2 the lowest. The difference between the $SO_2$ mixing ratios in the different schemes is greatest in January/December and

'51    lowest in June, both in the Central North Atlantic and the Southern Ocean. This pattern is similar for sulfate concentrations in

'52    the Southern Ocean, while sulfate in the North Atlantic is not affected by the different chemical schemes, resulting in similar

'53    concentrations for all simulations (due to the large contribution of anthropogenic sources). Additionally, here, sulfate

'54    concentration does not follow the same seasonal pattern as $SO_2$, contrary to the Southern Ocean, where anthropogenic

'55    emissions are minimal. In the North Atlantic, the maximum $SO_2$ and sulfate levels are reached close to the surface (**Figure**

'56    **15c,d**), tied closely to the fact that the major emissions – shipping and industry – are injected near the surface). $SO_2$ is depleted

'57    quickly in the boundary layer (similar to HPMTF in **Figure 11**), while sulfate concentrations decrease more slowly with height,

'58    owing to longer timescales for secondary production from intermediate lifetime DMS oxidation products. In the Southern

'59    Ocean however, the maximum $SO_2$ concentration is only reached at ~2 km in CS2 and ~3 km in CS2-HPMTF and ST. The

'60    opposite pattern is observed for the annual mean maximum sulfate concentration by altitude: 1.1 km for ST, 2.4 km for CS-

'61    HPMTF and 5.2 km for CS2. This can affect the climate response to DMS emissions because radiative forcing is sensitive to



'62    the altitude of aerosols (Krishnamohan et al., 2019). Ranjithkumar et al. (2021) also assessed the ability of UKCA to simulate

'63    $SO_2$ compared with ATom measurements. In their study they used the Lana et al. (2011) emissions and found that reducing

'64    the scaling factor to that used by Mulcahy et al. (2018), amongst other changes (cloud pH and aerosol microphysical process

'65    changes) gave them the best fit to observations.

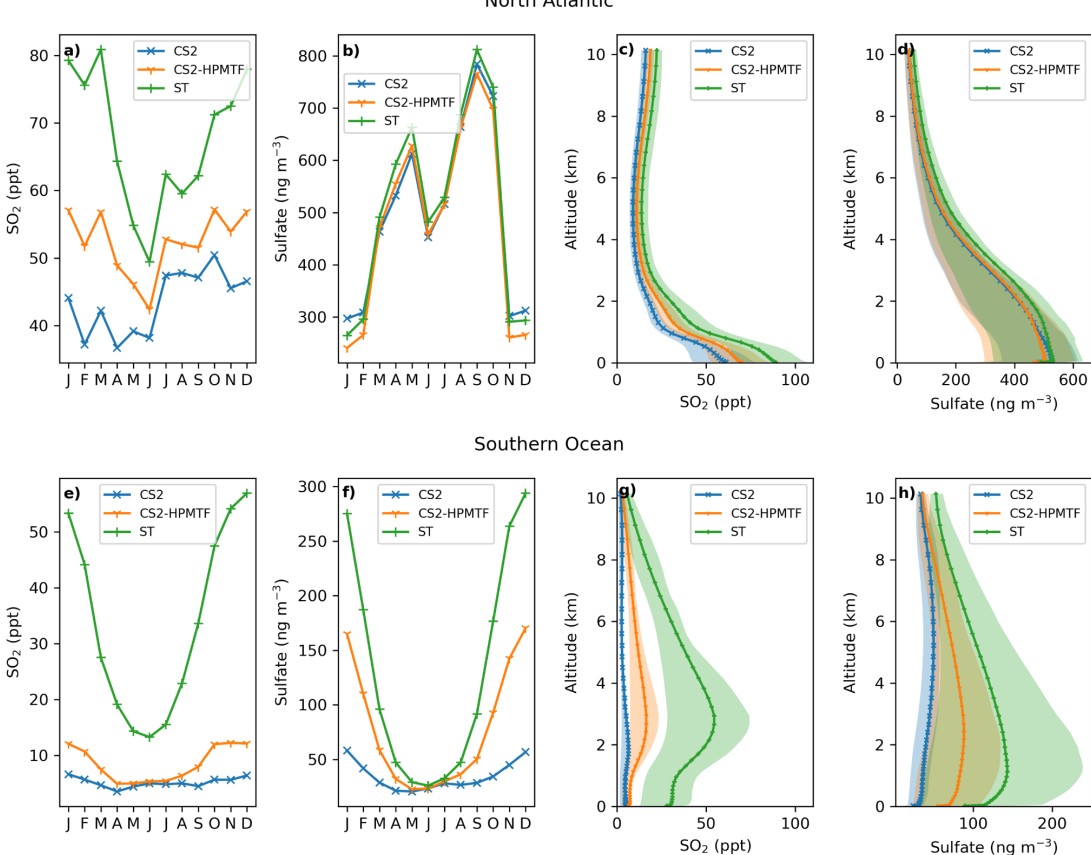

'66

'67    **Figure 15:** Monthly mean **(a)** $SO_2$ mixing ratios **(b)** sulfate concentration in the lower troposphere (< 2 km) and the annual

'68    mean vertical distribution of **c)** $SO_2$ and **d)** sulfate concentration in the Central North Atlantic (30-50°E, 20-45°N). The

'69    envelopes represent the interquartile range of the measurements. **e) - h)** the equivalent for the Southern Ocean (50-70°S).

'70

'71

'72    **Comparison to observed $SO_2$ and sulfate**

'73    **Figure 16a** shows the monthly means of observed non-sea-salt sulfate (nss-sulfate) concentration at Dumont d'Urville station

'74    (66°S, 140°E) between 1991 and 1995 (Minikin et al., 1998) and compares it to the sulfate concentration in the three different

'75    UKCA model runs. The seasonal changes in sulfate concentrations are reproduced by CS2-HPMTF and ST, but not by CS2.





'76   From April to September all three runs match the observations adequately well. Earlier in the year, the results from the ST run

'77   match the observations best, while later in the year CS2-HPMTF reproduces the measurements better.

'78

'79   **Figure 16b,c** show $SO_2$ measurements along the ATom-4 flight path in comparison with the modelled $SO_2$ concentrations. In

'80   the boundary layer, all runs over-predict $SO_2$ in comparison to the ATom-4 data **(Figure 16b)**. In addition to wet and dry

'81   deposition (Faloona 2009; Ranjithkumar et al., 2021), vertical mixing has been identified as a major source of uncertainty in

'82   models (Gerbig et al., 2008) and could provide an explanation for the mismatch between the simulation results and

'83   observations. At altitudes above 1.8 km, CS2-HPMTF is able to reflect $SO_2$ concentrations better than the other schemes.

'84   Above 9 km, the simulations underestimate $SO_2$, potentially indicating issues with convective transport. Overall, in the ATom-

'85   4 observations, $SO_2$ stays broadly constant with altitude, suggesting significant secondary sources or efficient vertical transport,

'86   while in the simulations it decreases. Additionally, the interquartile ranges of the concentrations in each bin are bigger,

'87   indicating a greater variance of model results than measured values. Overall, the mean $SO_2$ concentrations by the models in

'88   each latitude bin predict the mean observation values well **(Figure 16c)**. However, the variation of values is again greater in

'89   the model, especially at low latitudes.  The underestimation of $SO_2$ at 70°N could be due to an underestimation of the influence

'90   of anthropogenic $SO_2$ emissions or unrealistic deposition of $SO_2$ (Hardacre et al., 2021). Alternatively, the $SO_2$ production

'91   from DMS might be too slow still.

'92

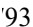

'93



'94

'95 **Figure 16: a)** Comparison of nss-sulfate concentration at the Dumont d'Urville Station (66°S, 140°E) at the coast of Antarctica.

'96 The observational data stems from Minikin et al. (1998) and represents the monthly mean concentrations and their standard

'97 deviations for the years 1991-1995. **(b)** Vertically binned (500 m) and **(c)** latitudinally binned (20°) median $SO_2$ mixing ratio

'98 along the ATom-4 flight path. The envelopes represent the interquartile range of the measurements and the respective model

'99 results while the numbers on the side/on top give the number of measurements in the respective bin.

'00

'01

'02 **4.3 Sensitivity runs**

'03 To improve our understanding of the variability of the model results, based on the uncertainties of HPMTF formation and loss,

'04 three sensitivity runs were conducted (CS2-HPMTF-CLD, CS2-HPMTF-FL, CS2-HPMTF-FP, **Table 1**). Loss of HPMTF to



clouds was proposed to be a major loss pathway by Veres et al. (2020) and Vermeuel et al. (2020). CS2-HPMTF-CLD adds cloud and aqueous uptake of HPMTF with a reactive uptake coefficient, γ, of 0.01, used in the study by Novak et al. (2021). Jernigan et al. (2022) recently established a rate constant for oxidation of HPMTF by OH as 1.4 (0.27–2.4) $\times 10^{-11}$ cm$^3$ molecule$^{-1}$ s$^{-1}$ through constrained chamber modelling using a rate constant for the formation of HPMTF as 0.1 s$^{-1}$. Further laboratory studies would be helpful in constraining this rate constant. Vermeuel et al. (2020) found the theoretically calculated rate constant $1.4 \times 10^{-12}$ cm$^3$ molecule$^{-1}$ s$^{-1}$ by Wu et al. (2015) too slow and proposed a rate constant of $1.11 \times 10^{-11}$ cm$^3$ molecule$^{-1}$ s$^{-1}$ instead, based on structurally similar molecules and modelling of their ground-based observations, similar to what we used in CS2-HPMTF. They recommend an upper limit of $5.1 \times 10^{-11}$ cm$^3$ molecule$^{-1}$ s$^{-1}$ for the HPMTF+OH rate constant. Khan et al. (2021) and Novak et al. (2021) use $5.5 \times 10^{-11}$ cm$^3$ molecule$^{-1}$ s$^{-1}$ for sensitivity tests, which was also employed in CS2-HPMTF-FL. Further, the study by Ye et al. (2021) looked at the uncertainty of the HPMTF isomerization rate. They estimate the isomerization rate constant as 0.09 s$^{-1}$ (0.03-0.3 s$^{-1}$, $1\sigma_g$ geometric standard deviation at 293 K). Veres et al. (2020) are on the lower end of this range (0.041 s$^{-1}$) and Berndt et al. (2019) at the higher end (0.23 s$^{-1}$). The CS2-HPMTF-FP simulation scales the rate constant of Veres et al. (2020) by a factor of 5 to match Berndt's measurements at 295 K to examine the effects of higher HPMTF production. This rate constant was also used by Wollesen de Jonge et al. (2021) in their study. The annual mean of global tropospheric burdens of relevant species in these sensitivity runs are compared in **Table 4.**

**Table 4:** Global annual mean tropospheric burdens of atmospheric sulfur species in UKCA base and sensitivity runs (first half of the table) and comparison to literature values (second half of the table, same acronyms as in Section 3)

| Run | HPMTF burden (Gg S) | SO₂ burden (Gg S) | Sulfate burden (Gg S) |
|---|---|---|---|
| CS2 | - | 370.1 | 582.3 |
| ST | - | 469.7 | 635.9 |
| CS2-HPMTF | 23.7 | 390.7 | 604.0 |
| CS2-HPMTF-CLD | 2.6 | 367.3 | 591.2 |
| CS2-HPMTF-FL | 8.9 | 392.6 | 605.6 |
| CS2-HPMTF-FP | 26.5 | 389.6 | 601.5 |
| FG$^\otimes$ (similar to CS2-HPMTF) | 18 | 365 | 582 |
| NV Base 1$^\oplus$ (similar to CS2-HPMTF) | 18.8 | 189.0 | 526.7 |
| NV Test 3$^\oplus$ (similar to CS2-HPMTF-CLD) | 0.7 | 180.2 | 550.7 |
| KH NEW_CHEM1$^\varnothing$ (similar to CS2-HPMTF, with photolysis of HPMTF) | 15.1 | - | - |
| KH NEW_CHEM2$^\varnothing$ (similar to CS2-HPMTF-FL) | 6.1 | - | - |

$^\otimes$Fung et al., 2021; $^\oplus$Novak et al., 2021; $^\varnothing$ Khan et al., 2021.

 

### 4.3.1 HPMTF

The HPMTF burden varies between 2.6 and 26.5 Gg S among the sensitivity runs (**Table 4**). Compared to CS2-HPMTF, faster OH oxidation reduces the HPMTF burden by -62% to 8.9 Gg S, while the addition of cloud and aqueous uptake to the scheme reduces it by -91% to only 2.6 Gg S. Yet, a factor of 5 higher production rate constant of HPMTF only leads to a 12% increase of HPMTF burden to 26.5 Gg S; suggesting that the steady-state distribution of HPMTF is controlled by the loss rate, not the rate of production of HPMTF. With the isomerization rate constant recommended by Veres et al. (2020), 51% of DMS forms HPMTF (86% of MTMP); with the faster rate in CS2-HPMTF-FP it is 57% (96% of MTMP). Since the use of the isomerization rate from Veres et al. (2020) already outcompetes the bimolecular reactions of MTMP, scaling the A-factor does not have a significant effect on the HPMTF yield from DMS. Overall, it can be estimated that globally 50-60% of DMS forms HPMTF (however, if more DMS is oxidised through the addition channel by BrO or multiphase reactions, this ratio could be lower). Consequently, HPMTF formation seems to be well constrained and the major uncertainties lie with the loss of HPMTF, which warrant additional measurements.

Similar to **Figure 12**, the HPMTF:DMS ratio is used in **Figure 17** to compare the results of the sensitivity model runs with ATom-4 observations. In general, schemes with a higher production and slower loss of HPMTF match the observations better, however, they still underestimate the measured ratios. A comparison was made to HPMTF:DMS ratios measured with no clouds present. Under these clear-sky conditions, when cloud uptake of HPMTF should not play a role in the measurements, observed ratios were even higher, leading to a greater difference between model results (which include clouds) and observations.

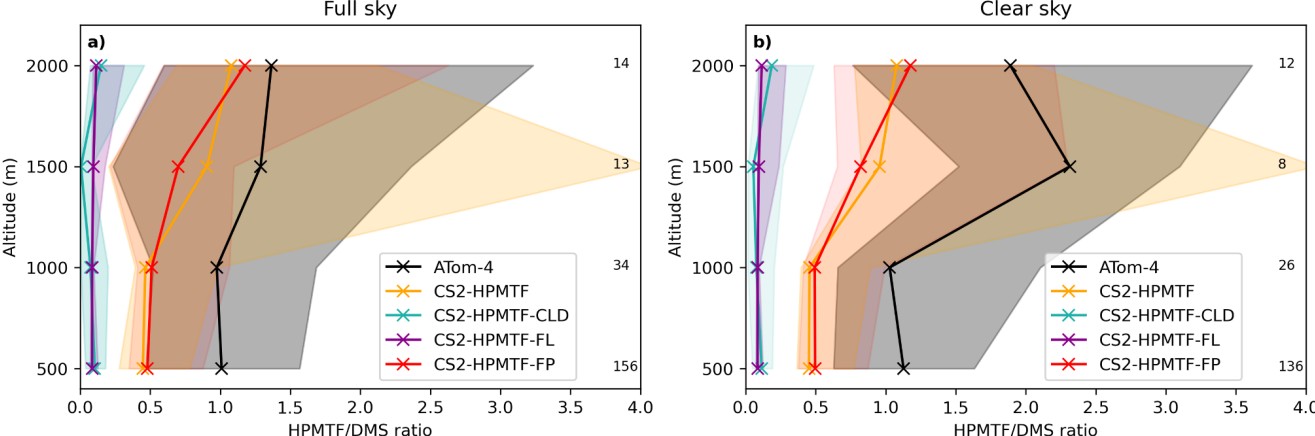

**Figure 17:** Vertically binned (500 m) median HPMTF/DMS ratio along the ATom-4 flight path for **a)** full sky and **b)** clear sky, where measurements made in clouds are omitted. The envelopes represent the interquartile range of the measurements and the respective model results, while values on the side give the number of measurements in the respective bin.





### 4.3.2 SO₂

The SO$_2$ burden varies between 367.3 Gg S in CS2-HPMTF-CLD and 392.6 GgS in CS2-HPMTF-FL, suggesting that the SO$_2$ burden is relatively unaffected by the chemical sensitives explored when compared with the much larger SO$_2$ burden simulated with ST (469.7 Gg S); mainly due to the 100% DMS-SO$_2$ yield (**Table 4**).

CS2-HPMTF, CS2-HPMTF-FL, and CS2-HPMTF-FP have a higher SO$_2$ burden than CS2 since the changes to the abstraction pathway (reaction 6a, 7c) and the addition of the isomerization pathway lead to more direct SO$_2$ production. Faster OH oxidation of HPMTF in CS2-HPMTF-FL reduces the amount of HPMTF deposited and therefore increases the SO$_2$ burden slightly (by 0.5%) compared to CS2-HPMTF. The faster production of HPMTF in CS2-HPMTF-FP reduces SO$_2$ burden marginally (-0.3%), due to more sulfur now being deposited as HPMTF or forming OCS. The addition of cloud and heterogeneous loss in CS2-HPMTF-CLD leads to immediate sulfate production instead of SO$_2$ formation, reducing the SO$_2$ burden by -6% compared to CS2-HPMTF, resulting in the lowest SO$_2$ burden in all runs considered.

### 4.4.3 Sulfate

In the sensitivity runs, the sulfate burdens are all higher than in the CS2 run (582.3 Gg S) and lower than in the ST run (635.9 Gg S). The variation by approximately 15 Gg S, from 591.2 Gg S in CS2-HPMTF-CLD to 605.6 Gg S in CS2-HPMTF-FL, is smaller than the variation in sulfate burden simulated by similar mechanistic sensitivity tests by Novak et al. (2021) (~24 Gg), suggesting some structural dependence on the results of the sensitivity tests (e.g., resolution, other model parameters). The sulfate burdens in CS2-HPMTF-FL and CS2-HPMTF-FP behave similarly to CS2-HPMTF. Since CS2-HPMTF-CLD added direct sulfate formation, a higher sulfate burden was expected. However, this was not seen in the experiments. Inspection of the sulfate aerosol distribution shows that CS2-HPMTF-CLD leads to an increase in the coarse mode sulfate and a concomitant reduction in sulfate aerosol lifetime (through an increase in wet deposition).



## 5 Discussion

The results described above demonstrate the global scale changes in the distribution of DMS and its oxidation products, through the incorporation of improved mechanistic updates into the UKCA model. Here we discuss our results in the context of the existing literature.

### 5.1 DMS

The DMS burden of 63-66 Gg S in this work is in good agreement with recent modelling studies (50 Gg S in Fung et al. (2022), 74 Gg S in Chen et al. (2018)). However, as shown in Section 4.1.1, the modelled DMS concentrations do not match observational measurements. One explanation could be underestimation of DMS oxidation. Here, only oxidation by OH and $NO_3$ is included. However, Fung et al. (2022), who include oxidation by BrO, $O_3$ and Cl (accounting in total for 20% of DMS depletion), also found that their model over-predicted DMS mixing ratios compared to the ATom-4 measurements. Inadequate representation of DMS concentrations in seawater and therefore emissions contribute to the largest uncertainties in the sulfur budget (Tesdal et al., 2016; Bock et al., 2021) and could explain most of the difference. Additionally, physical differences between model and observation, such as wind speed and temperature, and a poor space resolution of Whole Air Sampling might also play a role. Crucially, more long-term observations of DMS in the atmosphere are needed to complement works that have collated oceanic DMS observations (e.g., Lana et al., 2011).

Here, in all model runs 75% of DMS is oxidised by OH and 25% by $NO_3$. Other studies found global contributions of OH between 50-70% and $NO_3$ 15-30% (Boucher et al., 2003; Berglen et al., 2004; Breider et al., 2010; Khan et al., 2016; Chen et al., 2018; Fung et al., 2022). The lower contribution of OH oxidation to DMS removal is explained by the addition of other pathways, such as oxidation by BrO, Cl and multiphase reactions. Consequently, the lifetime of 1.5 days for DMS in this work is longer than some other studies including these reactions (e.g., 0.8 days in Fung et al. (2022) and 1.2 days in Chen et al. (2018)). Nonetheless, it is well within the range of 0.9 to 5 days (with a mean of 2 days) of the models examined in Faloona (2009).

### 5.2 HPMTF

In CS2-HPMTF 51% of DMS forms HPMTF. With a faster formation of HPMTF, found in laboratory experiments, this yield increases to 57% in our model. The yield could possibly be lower if other oxidation reactions of DMS are included that follow the OH addition pathway (multiphase reactions, oxidation by BrO). Veres et al. (2020) estimated that at least 30% of DMS was forming HPMTF, based on their observationally constrained model. Even though the rate of HPMTF formation is uncertain (Ye et al., 2021), it does not significantly affect the HPMTF yield from DMS, since it already outcompetes most





'05    other reactions of MTMP. For HPMTF formation, uncertainty seems to lie mainly at the branching ratio of the addition and

'06    the abstraction pathway of DMS. Indeed, the uncertainty in the HPMTF burden stems from the uncertainty of loss pathways

'07    and their respective contribution to HPMTF loss. Our model results agree well with the HPMTF burdens obtained by other

'08    global modelling studies, both in absolute values but also the relative changes we find in the sensitivity study (**Table 4**): In our

'09    sensitivity study a faster oxidation of HPMTF to OH lead to a decrease of 62% of the HPMTF burden, in Khan et al. (2021) it

'10    was 60%. In this work the addition of aqueous uptake of HPMTF reduced the burden by 91%, very similar to the reduction

'11    simulated in Novak et al. (2021) (96%).

'12

### 5.3 MSA

'14    The tropospheric MSA burden is 40 Gg S in CS2-HPMTF with a lifetime of 6 days. This falls within the range of 13-40 Gg S

'15    and a lifetime of 5-7 days found in previous model studies (Pham et al., 1995; Chin et al., 1996, 2000; Cosme et al., 2002;

'16    Hezel et al., 2011). However, newer studies include more multiphase processes and usually tend to have shorter lifetimes and

'17    lower MSA burdens. Both the scheme in Fung et al. (2022) and Chen et al. (2018), include the loss of MSA to aqueous OH

'18    oxidation, resulting in lifetimes of 0.6 days and 2.2 days and a burden of 8 Gg S and 20 Gg S, respectively.

### 5.4 SO$_2$ and Sulfate

'20    Comparing SO$_2$ and sulfate burdens with other modelling studies is more challenging, since those species can have other

'21    sources apart from DMS. That said, our SO$_2$ obtained in the various runs based on the CS2 scheme are comparable to Fung et

'22    al. (2022), while the ST burden is significantly higher. However, the SO$_2$ burden from Novak et al. (2021) is much lower. This

'23    difference cannot be explained solely by differences in the DMS oxidation mechanism; more likely, the difference is in

'24    anthropogenic SO$_2$ emissions.

'25    The sulfate burden in all our runs fall within the range found in other recent modelling studies (Chen et al. 2018; Novak et al.,

'26    2021; Fung et al., 2022). Considering the relative change due to the addition of the isomerization pathway, the increase in

'27    sulfate burden from CS2 to CS2-HPMTF is only 3.7% in our study, Fung et al. (2022) found an increase of 8.8%, when they

'28    added HPMTF chemistry. However, unlike their results, we find strong seasonality in the additional sulfate produced,

'29    especially in the Southern Hemisphere. The addition of cloud uptake and direct sulfate formation in CS2-HPMTF-CLD

'30    decreased the sulfate burden in our study by (-)2.2%, in Novak et al. (2021) this change in mechanism lead to an increase of

'31    sulfate by 4.5%.

'32

### 5.5 Comparison with BOXMOX results.

'34    In Section 3 and Section 4 we have shown the results of BOXMOX and UKCA simulations using different DMS mechanistic

'35    variants respectively. Whilst the same mechanistic variants have been assessed in both model setups, it is not possible to

'36    directly compare the results of the two sets of experiments because of the large differences in the model setups used. However,





some qualitative comparisons can be made. For MSA, Section 3.1 (Figure 2) suggests that the MSA simulated with CS2-
HPMTF should be much lower than CS2; as is calculated in Section 4.2.2 (a 70% reduction). For $SO_2$, both the BOXMOX
and UKCA results agree in the ordering of simulations, ST, CS2 and CS2-HPMTF; with ST simulating significantly more $SO_2$
than the other mechanisms. However, whereas BOXMOX simulations suggest that $H_2SO_4$ is predicted to be higher in CS2 and
CS2-HPMTF than ST, the UKCA model runs suggest that ST has the greatest burden of sulfate; highlighting the complexity
of making inference on aerosols from gas phase precursors in box model studies.

## 6 Conclusion

DMS remains an important molecule in our understanding of the background aerosol budget and the uncertainty of aerosols
to climate change (Carslaw et al., 2013). In this study we have used a combination of box modelling experiments and global
3D model experiments to explore the sensitivities of the DMS oxidation mechanism in the UKCA model. This work has
delivered a new DMS oxidation mechanism for use within the CRI-Strat framework of UKCA (Archer-Nicholls et al., 2021;
Weber et al., 2021), which is a significant advancement and improvement over the mechanism used in CMIP6 studies
(Archibald et al., 2020). Our new DMS mechanism includes many of the recently discovered and proposed oxidation pathways
for DMS and through the series of experiments we have performed, we have been able to benchmark this scheme against other
recently reported schemes in the literature. Whilst future studies building on the ever expanding database of laboratory studies
(e.g., Ye et al., 2021; Jernigan et al., 2022) are required to refine the DMS oxidation mechanism further, with the current
availability of observational data, it is not possible to fully constrain the uncertain parameters in the DMS oxidation
mechanism. Hence there is a priority for more observational based studies that combine ship, ground-based and aircraft
platforms optimally. Fung et al. (2021) have shown that there are consequences for radiative forcing by updating the DMS
mechanism in the CESM model, and follow up work will investigate these changes with UKCA.

This study adds to the few other mechanism intercomparisons that exist in the literature, spanning back more than 25 years
(Capaldo and Pandis 1997; Karl et al., 2007). Similar to these other studies we find that MSA is particularly uncertain when it
comes to the results obtained using the range of mechanisms that we investigated. Further work should explicitly focus on
reducing uncertainty in the MSA budget in the atmosphere, especially given its potential importance in reconstructing paleo-
sea ice (Thomas et al., 2019).

In many ways, the recent advances in DMS oxidation chemistry are similar to isoprene chemistry, where over a decade ago
the discovery of uni-molecular isomerisation reactions resulted in a step-change in our understanding of isoprene. As with
isoprene, ever more complex and faithful descriptions of DMS chemistry will be delivered over the coming years. But the
biggest challenge (as for isoprene) will remain in reducing and accurately distilling down this complex chemistry for use in





global model studies, and in characterising the sources of DMS into the atmosphere (which for isoprene have only recently
been possibly directly e.g., Wells et al., 2020).

**Acknowledgements**
BAC thanks the Studienstiftung des Deutschen Volkes for financial support. We would like to thank NERC through the ACSIS
(NE/N018001/1) and CARES projects for funding (NE/W009412/1). We would like to thank the UK Met Office JWCRP and
Clean Air programmes and the National Centre for Atmospheric Science for funding the development of the UKCA model.
LER acknowledges support from the Deep South National Science Challenge (contract C01X1901). ATA thanks the
University of Canterbury Erskine Programme. This work used Monsoon2, a collaborative high-performance computing facility
funded by the Met Office and the Natural Environment Research Council. This work used JASMIN, the UK collaborative data
analysis facility.

**Competing interests**
The authors declare no competing interests.





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
