# Peer review of "Development, intercomparison and evaluation of an improved"

_Atmospheric Chemistry and Physics, 2023_

## Author Comment (AC1)

Replies to reviews of "*Development, intercomparison and evaluation of an improved mechanism for the oxidation of dimethyl sulfide in the UKCA model*" by Cala et al.

Dear Prof. Orlando,

Firstly, our sincere apologies for the tardiness of our reply. The lead author of the study has moved on to a new position just before we submitted the manuscript and was then unavailable for a large period of time on a research cruise. Coupled with teaching and other administrative tasks I'm afraid that this took some time to pull together. But we have, herewith, replied to all the comments from the two anonymous reviewers and following their joint recommendation edited down the manuscript to reduce the length. We have however kept a lot of that original material in the now expanded supplementary information (SI).

The original comments from reviewer/referee 1 are given in red text below, and reviewer/referee 2 in blue text; black text is used for our responses.

Referee 1:

The authors develop a new DMS mechanism based on the literature before testing it with box model simulations and on a global scale. In the box model, they evaluate two versions of their new model against older models of low and medium complexity, examining both timeseries and sensitivity to temperature. They then adapt four recent literature mechanisms to the box model, and compare timeseries and temperature sensitivities. Next, they run their full new mechanism, the older medium complexity mechanism, and the simple mechanism in a global model. These are compared to each other and to assorted measurements. Finally, they run the global model with additional mechanisms exploring the addition of cloud loss, the addition of a faster HPMTF loss rate, and the addition of faster HPMTF production.

Key conclusions: The most novel aspect of this paper is the comparison between contemporary DMS oxidation mechanisms, which as the authors note has not been done since 2007. While the conclusions from this section could be stronger, these data would be of interest to the research community. Of secondary importance: the authors provide interesting discussion of some of the sensitivities and uncertainties in variations of their mechanism. They also demonstrate that their new mechanism represents an improvement over previous schemes for the UKCA model.

Major comments:

The paper is unnecessarily long and does not clearly frame the authors' conclusions. **I would suggest halving the length and centering the work and analysis around the comparisons with literature mechanisms.** This could primarily focus on the box model results while briefly touching on the global modeling, which seems to show similar sulfur burdens and sensitivities compared to the literature. As the manuscript currently stands, the authors carefully explain so many observations from their model perturbations that **it is difficult for the reader to take home a clear message. Ideally, these observations should be condensed and focused towards a central idea.**

We thank the reviewer for their time in reading and commenting on our manuscript. The manuscript, we agree, is on the long side and we take on board the reviewers comments that a reduction in length and a re-framing to focus on uncertainties will lead to a paper that will better serve the community. To this end we have made some significant editorial changes to the manuscript that can be seen in the tracked changes version. Before making these changes we have addressed the point-by-point comments of both reviewers – and added our responses to these below.

The paper also **insufficiently addresses the authors decision to leave out halogen and multiphase/aqueous chemistry**. The authors state that "the contribution [of BrO and Cl] is either

negligible or there is large uncertainty," citing only two papers. While prior work is variable (see discussions in Fung et al., 2022 and Hoffmann et al., 2016), it largely suggests that **DMS + halogen chemistry is an important sink on the order of 10% o**r more and its omission should at least be much more clearly explained. Considering that it could be reasonably represented with only one or two reactions, it is not clear why this is left out. Multiphase/aqueous chemistry, which has previously been shown to be important, is also omitted. **Since this typically requires different treatment in the model, its omission is more understandable but should be more clearly discussed.** Due to the omission of halogen and aqueous chemistry, discussions of the relative importance of different oxidants do not seem worthwhile.

We thank the reviewer for their comment about our omission of halogen and multiphase chemistry. We address this comment (and the related comment by reviewer 2) below.

1) Omission of halogen chemistry. Whilst it is well known that DMS undergoes reactions with halogens the oxidation by these species either leads to the production of DMSO or MTMP. Omitting halogens is akin to having biases in [OH] and [NO$_3$] (both of which have uncertainties on the order of 10%); species who we do simulate oxidising DMS in our model. Our study, motivated by synthesising past and recent work, is more concerned with the fate of these two primary oxidation products (DMSO and MTMP). Our omission of halogen chemistry has little impact on the aims of our study, which are to address the uncertainty in the oxidation of these primary products. We will make this clearer in the manuscript.

2) Omission of multiphase/aqueous chemistry. We actually do address multiphase chemistry by looking at the fate of HPMTF in clouds, but we agree with the reviewer that more details can be added to the paper to explain i) the importance of multiphase oxidation ii) our choice to omit it. In terms of ii) this is down to the wider issue of adding in aqueous phase chemistry in UKCA. UKCA deals with the aqueous oxidation of S(IV) to S(VI) through SO$_2$ oxidation. However, the module that this chemistry is written in was developed exclusively for the aerosol component of UKCA (GLOMAP-Mode). As such it is very hard-wired and hard to modify. We do wish to adjust this module to a more flexible format and in the future we plan a comprehensive study on multiphase DMS chemistry.

Specific Comments:

Line 18: The authors imply mechanistic uncertainty is the main reason that mechanisms in global models are oversimplified. I believe this is more a result of attempts to keep the model efficient since sophisticated mechanisms for DMS have been around for at least twenty years.

We partially disagree on this point and feel it is the issue of mechanistic uncertainty that has led global modellers to be very succinct with the DMS oxidation mechanisms over time. Modellers will prefer it if you tell them there are fewer reactions needed to model the system at study in order to keep the model as simple as possible. Indeed, this is what was done in Faloona (2009) when focusing on the yield of SO2 from DMS. But we don't agree that this is **the only reason** why models have kept with very simplified DMS chemistry for 27+ years. We believe this is because of the high mechanistic uncertainty in DMS oxidation that has meant that rather than expand the chemistry and include lots of uncertain reactions, the community has favoured an approach of "this is not perfect but it will do". We also don't believe we have any sophisticated mechanisms for DMS, at least not to the same extent that we do for isoprene or methane. The canon of mechanistic literature is sparsely filled with sulfur studies, which makes understanding the uncertainty of DMS chemistry hard (fewer studies does not enable greater certainty). Sulfur species make up less than 0.3% of the species reviewed in McGillen et al. (2020); the most comprehensive primary database on gas-phase kinetic data published in the peer-reviewed literature. As a result the few large DMS mechanisms are sub-sampling the amount of chemical space available during the oxidation of DMS and are largely drawn from carbon based structure activity relationships when moving away from the primary oxidation products/reactions. One of the original attempts at a comprehensive

DMS mechanism was by Yin et al. (1990), with many of the rate constants and reactions based on an educated guess by the authors. Barnes et al. (2006) were able to provide some updated reviewed laboratory evidence for many more reactions than Yin et al. (1990) but also highlighted a number of key reactions were uncertain.

The MCMv3.1 DMS mechanism is widely used in the literature as the most comprehensive DMS gas phase mechanism but was not formally developed (optomised) against chamber experiments (post hoc, several studies have made use of it and highlighted skills and weaknesses of the MCM e.g., Ye et al., 2022; Jernigan et al., 2022). We are in a position now where global modellers have opted for parsimony as the spread in more complex mechanisms is large and has not been reduced for decades. Faloona (2009) reviewed Karl et al. (2007) who had reviewed several schemes (7) of which one of the simplest, Chin et al. (1996), "performed extremely well with only 6 reactions in comparison to many of the more complex models." Faloona then recommended that this reduced mechanism be used in sulfur modelling, and so most models have followed suit.

We will make our point clearer in the text to expand on why we believe it is mechanistic uncertainty that has led to the adoption of very simple treatment as well as modellers wanting to keep things simple.

Line 57: Consider additionally citing Fung et al., 2022.

Done.

Line 67: Reference S1.4.1.

Done.

Line 86: Cite other HPMTF yields (ex. Novak et al., 2021 à 46%, Fung et al., 2022 à 33%).

Done.

Line 89: Consider citing Ye et al., 2022. This paper includes a more up-to-date figure that demonstrates the uncertainty the isomerization rate constant.

We feel the text correctly describes the work of Ye et al. (2022) but we have added in the Assaf et al. (2023) study.

Line 111 or thereabouts: Similar sensitivity studies have been done by Fung et al., 2022 and possibly others. How does this work differ or add to this literature?

We have added text to contextualise that we are complementing these studies with a structurally different model.

Line 122: Consider specifying the NO/NO2 ratio. [NO] is of course quite important for MTMP fate.

This ratio can be clearly calculated from Table S1 unless the reviewer means the ratio during the box modelling? If it's the latter we don't think that is so useful to plot – and given we were instructed to cut material we don't think that is useful to add either.

Line 209: Why is MSA not considered here?

It's a great question! On hindsight we would have but we started out wanting to focus on the recently identified HMPTF and the more classic oxidation products of SO2 and sulfate as these connected to the climate modelling work done in CMIP6. Future work will focus on MSA.

Line 297: Measurement of HPMTF + OH (Ye et al., 2022) should be mentioned in this section.

Done.

Line 331: It is clear that a steady state is not achieved for MSA and SO2 in the time shown in Figure 2.

We have changed the text to quasi-steady state.

Line 371: It is not clear to me that in-depth comparison of MSA for ST is relevant since MSA is barely treated by these schemes.

We disagree and think it's important for context.

Line 468: This is surprising. Why does this significant loss not affect the diel profile?

We agree it is a surprising result from Khan et al but to be clear we don't include this photolysis pathway as in our modelling we didn't see it as being important in the box model. We note an unpublished presentation by the Julich group (Atmospheric Chemical Mechanisms conference, 2022) found some experimental evidence to support the Khan et al finding, which will provide an interesting case for follow-up work.

Line 476: HPMTF may deserve more discussion here. A significant factor in the daytime decrease at the higher NOx level may be that production of HPMTF slows down due to competition from MTMP + NO. Is OH significantly different between the 10 ppt and 100 ppt NOx simulations?

During the day [NO] is very low, even at higher NOx, so we ascribe the main driver for the diel variation down to the rate of DMS oxidation, as discussed in the text.

Line 520: What fraction of H2SO4 in your model is produced by SO2 vs CH3SO-3?

Thanks for the question, 41.5% of the $H_2SO_4$ in the BOXMOX simulations with the CS2-HPMTF mechanism is produced by $CH_3SO_3$.

Line 567: The SO2 temperature sensitivity could be more clearly explained.

We think that this section is clear and are unsure what the reviewer is suggesting.

Line 578: Conclusions here are quite vague. The authors could for example more clearly discuss the agreements and disagreements of the different mechanisms and anticipate the impacts of changing global temperatures on DMS chemistry.

We have expanded on this section.

Line 593: ratios are > 100 ppt. As written, this statement does not really make sense.

We think the sentence makes sense as the mixing ratio (pmol/mol) is relatively high but variable: between 100 and greater than 300 pptv.

Section 4.1.1 The authors note that the model is biased very high compared to measurements. It may be worth noting that this has been seen in other modeling studies as well.

We added a ref to Fung et al who also found this.

Sections 4.1.1, 4.2.1, 4.2.3: What is the impact of comparing the model with measurements from different years? This seems particularly notable for SO2 which has major anthropogenic sources.

This is a good point and something we can't look into at present. The main species we rely on older data for is DMS and we have tried to use "climatological" data which should be robust to model year as well as data from years we are explicitly simulating. We don't think this is a major issue but we would need to run more simulations (beyond the scope of the project) to confirm this.

Line 662: It is clear that the lack of an HO2 pathway in the CS2 mechanism is an obvious flaw for modeling MTMP over the ocean. This analysis doesn't emphasize this fact.

Agreed. We added "This is clearly wrong and a failure of the CS2 scheme."

Figure 16 c. Due to anthropogenic sulfur and differences in modeled vs measured years, it isn't clear how relevant this comparison is.

Dumont d'Urville Station (used in panel a) is in Antarctica where we don't expect large anthropogenic signals. The other panels show data that were collected in the time window the model was run for. We haven't done an apples to apples comparison but we feel that's not necessary for these analyses.

Figure 13.a.: Is the heterogeneity in MSA greater than for DMS? This seems interesting.

It's a great question, we don't have a good way of measuring it but playing around with the colour scales makes it clear that on a linear scaling DMS is more heterogeneous than MSA. We intend to keep the current linear colour scales to keep this clearer.

Line 809: See OH + HPMTF rate measured in Ye et al., 2022.

Thanks, we added the following:

" Ye et al. (2022) also measured the rate constant for this reaction. In their study they derived a rate constant of $2.1 \times 10^{-11}$ $cm^3$ molecule$^{-1}$ s$^{-1}$ and an isomerization rate constant, $k_{isom}$, of $0.13 \pm 0.03$ s$^{-1}$ at 295 K. Whilst, further laboratory studies would be helpful in constraining the rate constant for OH + HPMTF, we recommend future work go into constraining the products of this reaction"

Line 841: If this is the case, why is HPMTF/DMS for CS2-HPMTF-CLD still so low in panel b?

Thanks for this, the text was not very clear and we have amended it to make it clearer that the model results are the same in panels a and b, it's just the observations that we separated out into all-sky and clear-sky as the model diagnostics were not output for us to do the same with the model data.

Line 851: What fraction of SO2 is actually from DMS oxidation?

In the ST runs we performed for CMIP6 we calculate that the flux of DMS oxidation forming SO2 is ~ 16 Tg(S) yr$^{-1}$ (i.e. the same as the DMS emissions). This makes sense as the $SO_2$ yield from DMS was 100%. The direct anthropogenic emissions are ~ 60 Tg(S) yr$^{-1}$ in the present day, making the fraction of SO2 that is actually from DMS oxidation 21% (Hardacre et al., 2022). In the CS runs (the base runs used in the present study) we calculated a much lower contribution of DMS to the $SO_2$ production flux; 29% yield of $SO_2$ from DMS and a fraction of 7% SO2 from DMS. The CS2-H run, with a 65% yield of $SO_2$ from DMS oxidation, lead to a fraction of 14% $SO_2$ coming from DMS.

Line 903: See Line 86 comment.

Done.

Line 904 & 907: Fung et al., 2022 also reported that HPMTF burden is not sensitive to the isomerization rate constant and quite sensitive to cloud uptake.

Added this in, thanks.

Section 5.5 is separated from the data by ~20 pages so it doesn't feel that relevant by the time you get to it. I think it is possible to draw larger conclusions from these data.

We have added in a summary section in the BOXMOX modelling work to amend this issue.

Technical Corrections:

Typos and run-on sentences found in lines 53, 208, 435, and 727. Typo in Table 1.
Done, thanks!

References:
Barnes, I., Hjorth, J. and Mihalopoulos, N., 2006. Dimethyl sulfide and dimethyl sulfoxide and their oxidation in the atmosphere. *Chemical reviews*, *106*(3), pp.940-975.

Chin, M., Jacob, D. J., Gardner, G. M., Foreman-Fowler, M. S., Spiro, P. A., and Savoie, D. L. (1996), A global three-dimensional model of tropospheric sulfate, *J. Geophys. Res.*, 101( D13), 18667– 18690, doi:10.1029/96JD01221.

Faloona, I., 2009. Sulfur processing in the marine atmospheric boundary layer: A review and critical assessment of modeling uncertainties. *Atmospheric Environment*, *43*(18), pp.2841-2854.

Karl, M., Gross, A., Leck, C. and Pirjola, L., 2007. Intercomparison of dimethylsulfide oxidation mechanisms for the marine boundary layer: Gaseous and particulate sulfur constituents. *Journal of Geophysical Research: Atmospheres*, *112*(D15).

McGillen, M. R., Carter, W. P. L., Mellouki, A., Orlando, J. J., Picquet-Varrault, B., and Wallington, T. J.: Database for the kinetics of the gas-phase atmospheric reactions of organic compounds, Earth Syst. Sci. Data, 12, 1203–1216, https://doi.org/10.5194/essd-12-1203-2020, 2020.

Yin, F., Grosjean, D. & Seinfeld, J.H. Photooxidation of dimethyl sulfide and dimethyl disulfide. I: Mechanism development. *J Atmos Chem* 11, 309–364 (1990). https://doi.org/10.1007/BF00053780

Referee 2
Dimethyl sulfide (DMS) is a sulfur containing volatile compound emitted from the ocean. It is oxidized in the atmosphere and form a range of compounds including sulfate. DMS is important in the global sulfur budget and the oxidation compounds contribute to the formation of efficient cloud condensation nuclei. During recent years, new knowledge on the oxidation mechanism of DMS has appeared and it is important that this knowledge is implemented in models. At the same time, models may help in elucidating where additional data or mechanistic insight is needed.

This manuscript describes an extensive effort to update the UKCA chemistry-climate model with a more detailed description of the atmospheric oxidation of DMS than in the current version.

While the authors should be complemented for their efforts and for implementing and comparing several chemical schemes, I fully second Reviewer 1 in that the manuscript is much too long. It is hard as a reader to get an overview of what the main findings of the work are. The comparison with the box-model of the different chemical mechanisms is very interesting and I think it could strengthen the paper if the authors based on this could come up with a list of concrete key problems to address in laboratory and field studies to help constrain models on DMS oxidation and fate of the oxidation products.

Below are some more detailed comments and suggestions. In particular, the last part of the paper on global model runs I find difficult to read and it has several sections, which do not really provide conclusions and could be shortened. Instead, the authors could expand on the observations and messages from the first part. Thus, my comments at this point are mainly to the first part of the manuscript.

Abstract: The abstract contains several abbreviations (ST, CMIP6,).

Fixed.

Line 31: "based on the observed DMS/HPMTF ratio" It should state where this ratio was observed.

Fixed.

Line 32: "with a significant divergence in the sensitivity of these products to temperature" – this should be reformulated – the products themselves are not sensitive to temperature – rates of formation or similar can be sensitive to temperature.

Done.

Line 56: "due to the uncertainty in DMS oxidation" – I suggest to say uncertainty about the kinetics and mechanism involved in the oxidation of DMS.

Done.

Line 61: were the initial conditions the same across the six different chemistry schemes? This should be stated.

Yes, added.

Equations R1-R4 are not balanced chemically  - should they not be?

No, they don't need to be as they illustrate what is in the model, which is not balanced (partly because things like H2O and CO2 are not considered as species that are solved for in the model).

I suggest to provide a table in the supporting material with the 19 reactions included in the CS2 model.

Done.

Page 3 In line 87 the authors mention kisom,1 – the rate constant of the first H-shift.  Later it is referred to as kisom– should be consistent.

We have added text to make this clearer.

Page 4 line 12: sensitivity studies with a slower loss, a faster production …" – it should be explained what the authors compare to – faster than what?

We have tidied this up.

Page 4 line 20: is there a reference for the values used in Table S1?

No, we designed much of this ourselves.

The reference Glasow and Crutzen 2004 is referenced multiple times but is missing in the reference list. I assume it is Glasow and Crutzen ACP 2004. Here the DMS emission rate in the remote marine boundary layer is given as only 2x 10^9 molecule cm-2 s-1 – why was a higher value used here (corresponding to Cape Grimm summer in Glasow and Crutzen 2004)?

Thanks for this, and yes, that's the paper we mean. We looked at the smaller open ocean emissions value but this led to much smaller levels of DMS being simulated, so we opted for the values stated in the manuscript – as you say, reflecting the summertime Cape Grim flux from von Glasow and Crutzen.

Page 5: the first paragraph is difficult to read. For example, the last sentence: "the data from day 7 and 8 of the runs was averaged to enable the effects of changes in the temperature on species concentration simulated in the box model to be calculated" I do not understand the meaning of this sentence – why is it necessary to average from two days to see and effect of temperature? Can the first six days not be used?

We have tidied up the language in this paragraph to make it clearer. We allowed the box model to run for six days to enable a quasi steady-state to be achieved for several species. We then averaged two days as the model was still at a quasi steady-state i.e. there were several species still undergoing some changes. This is always an issue with box modelling studies but we have shown both the time evolution plots and then the averaged data so that the reader can get a good impression of what we have done, which follows from previous work in the group.

Page 5 and 6 on the 3D simulations – there is information that is perhaps not so relevant here. I suggest to provide references and only focus on the DMS chemistry in this description – to shorten the text.

Done.

I suggest the authors add a short explanation what the nudging done is.

We included a long description of this which we have now moved to the SI to reflect your previous comment.

Figure 1 is quite central to follow the manuscript. I strongly suggest the authors add equation numbers to arrows corresponding to those in the tables. I suggest that the authors provide both the chemical formula and the abbreviation – e.g. for MSA the chemical formula is missing. For the reaction of MTMP with NO (2a) in the figure it also says MSP in Figure 1 on the right side of the arrow – I do not find that in Table 2?

Done.

Page 9 The authors should expand a bit on the reason why they do not consider oxidation by BrO and Cl.

We have added some text to this effect based on the more detailed response to reviewer 1.

Page 10: the authors note that the Henry's law constant they use for MSA is two orders of magnitude higher than that used by Wollesen de Jonge et al. – what was the motivation for this. I might have overlooked it but did the authors do a sensitivity study to see the impact of varying this Henry's law constant?

Thanks for this comment – this took a while for us to work out (the joys of units). No, we didn't look into the sensitivity of this and as we are talking about the wet removal we don't think that's in the scope of this work. The wet removal will be as affected by the choice of $H^{cp}$ as it will be by the clouds in the model – which we can't constrain. We used the Henry's law constant that was recommended by Rolf Sander in his compendium and have added that reference, too. We did make a mistake in comparing the value we used and the value from Wollesen de Jonge and we have made the difference (a factor of 10) clearer.

Tables 2 and 3: why are not all the chemical reactions chemically balanced ? They should be balanced so that it corresponds to the rate constants given.

It's common practice in modelling atmospheric chemistry that species whose tendencies are not solved for with a continuity equation (so called constant species) are not included in mechanisms. These tend to include things like $O_2$, $CO_2$ and $H_2O$ as products. The reactants are complete as the concentrations of the reactants are required to calculate the rates of change.

Reaction 2c and others – please explain the factor in bold that is multiplied with the rate constant.

This is just the branching ratio, as indicated with the % sign in Figure1.

Page 12 line 82: I believe it should be reaction 2d and not 2c in parenthesis.

Indeed, apologies and great spot! We have fixed it, thanks!

Table 3: for some of the rate constants it is stated that they are from this work – how were they obtained? This should be stated in the heading or with a reference to the section where it is explained.

Thanks. We have added the link to the section where they are described.

3.1: In the text it says "Comparing CS2-HPMTF and CS2-UPD-DMS we can see that this is due to reaction 7c" – please explain in more detail how this can be seen.

Thanks. We have expanded the text to make this clearer:

"Comparing CS2-HPMTF and CS2-UPD-DMS, we can see that this pattern (increased $SO_2$ and decreased MSA) is due to reaction 7c, which directly forms $SO_2$ and suppresses $CH_3SO_2$, consequently lowering MSA formation. The $SO_2$ concentration is lower in CS2-HPMTF compared to CS2-UPD-DMS because the addition of HPMTF produces OCS which acts as a long-lived sulfur reservoir."

When talking about lifetimes please give an approximate value, e.g. line 46 page 14: SO2 is a relatively long-lived species – here the lifetime should be given.

Thanks, done.

 the temperature sensitivity between 270 and 290 K is attributed to difference in the rate constant of DMS oxidation through the OH-addition channel. Did the authors test this statement by running the models with the same rate constant for this reaction?

That's a great question, we didn't do that but as we use a consistent inorganic chemistry and we checked that things like OH are very similar between the model runs we felt we could safely assess this by just comparing the rate constants at 298 K and 1 atm. These are inconsistent across mechanisms hence we attribute this as the initiator of the divergence.

Can the authors show how the model develops and when steady state in the model is reached? For example plotting the DMS, MSA, SO2 and H2SO4 concentrations versus time in one panel and the temperature versus time in panel below on the same time scale – this would help the reader get an idea of how the model develops towards steady state.

In part we have addressed this comment related to comment re Line 331 by reviewer 1 by adding to the text that the model results are at quasi steady-state. The time evolution of the BOXMOX simulations are shown in Figures 2, 4 and 5. One can see that many species do reach a steady state within a few days (2-3 days) but others, such as H2SO4 and OCS don't as they have either very long lifetimes (OCS) or sinks that we did not simulate (H2SO4 has an important depositional sink and aerosol sink we did not model).

 Perhaps I misunderstand something, but for DMSO the authors discuss deposition as a significant sink but deposition is not included in the box-model? (same on page 21 where deposition of side products.

That's right, we didn't include deposition as a sink but we wanted to make clear that for the global budget it is an important sink.

 I miss a paragraph about what the main conclusions in 3.1.1 are.

Section 3.1.1 refers to the time series analysis on page 15. We think the reviewer means Section 3.2.1 on page 22? We have added the following summary to the revised manuscript.

*"To summarise, the intercomparison of recent gas-phase DMS oxidation mechanisms complements and extends earlier studies on DMS (Karl et al., 2007). Recent gas-phase DMS oxidation schemes used in modelling studies lead to a wide range in results of key DMS oxidation products, with moderate NOx levels (~ 0.1 ppb) leading to greater divergence than low NOx levels (~ 10 of ppt). A similar situation was found for isoprene by Archibald et al. (2010) and significant efforts have been employed to improve our understanding of isoprene oxidation through theoretical and laboratory experiments (e.g., Jenkin et al., 2015; Wennberg et al., 2018). We now focus on the role of temperature on the divergences seen thus far."*

Minor:

 something is wrong with the year in the reference 20114
Thanks! Fixed to 2004!

New references added:
Lee, C., Martin, R. V., van Donkelaar, A., Lee, H., Dickerson, R. R., Hains, J. C., Krotkov, N., Richter, A., Vinnikov, K., and Schwab, J. J.: SO2 emissions and lifetimes: Estimates from inverse modeling using in situ and global, space-based (SCIAMACHY and OMI) observations, J. Geophys. Res.-Atmos., 116, D06304, https://doi.org/10.1029/2010JD014758, 2011.

Jenkin, M. E., Young, J. C., and Rickard, A. R.: The MCM v3.3.1 degradation scheme for isoprene, Atmos. Chem. Phys., 15, 11433–11459, https://doi.org/10.5194/acp-15-11433-2015, 2015.

Wennberg, P. O., Bates, K. H., Crounse, J. D., Dodson, L. G., McVay, R. C., Mertens, L. A., Nguyen, T. B., Praske, E., Schwantes, R. H., Smarte, M. D., St Clair, J. M., Teng, A. P., Zhang, X., and Seinfeld, J. H.: Gas-Phase reactions of isoprene and its major oxidation products, Chem. Rev., 118, 3337–3390, https://doi.org/10.1021/acs.chemrev.7b00439, 2018.

---

## Author Response (AR2)

Dear Prof. Orlando,
Please accept further apologies for the time it's taken us to reply to the re-review of this paper and to address your editorial comments/suggestions.

We appreciate the referees comments that this is a publishable study but that it was too long – even after we expanded the SI. We have placed more material into the SI, including the original S2.2 describing how we put our new mechanism together. We hope that the paper is now shortened sufficiently to make it more widely readable. We have also added some very specific comments for the need for more laboratory studies on small organo-sulfur radicals that are urgently required to reduce uncertainties in mechanisms.

Comments from Anonymous Referee#2 in grey, replies in **black** and blue.
The manuscript is still a bit on the long side and a bit difficult to get the overview and find the key messages. In my previous review I wrote: "The comparison with the box-model of the different chemical mechanisms is very interesting and I think it could strengthen the paper if the authors based on this could come up with a list of concrete key problems to address in laboratory and field studies to help constrain models on DMS oxidation and fate of the oxidation products."

The current text about future lab and field work in the conclusion is very broad and general. As far as I can see, the authors have not added such as concrete list? I still think it would strengthen the paper if they could add such recommendations based on their model results.
Thanks for re-reading this and for your helpful comments. We agree about adding a more concrete list and have added such statements into the conclusions and abstract.

"Our results suggest that as a priority laboratory studies are performed that address 1) the uptake of HPMTF onto aerosol surfaces and the products of this reaction. 2) The kinetics and products of the following reactions: $CH_3SO_3$ decomposition; $CH_3S + O_2$; $CH_3SOO$ decomposition; $CH_3SO + O_3$."

Line 135-136: It should be stated that the authors chose the higher value for DMS emission in Glasow and Crutzen 2004.
Corrected.

Page 16 line 371 – 372 The authors refer to S1.3.1 in the supporting material – I cannot find such a table.
Sorry, this is a typo and refers to S1.4.1. We have corrected this.

Also the temperature sensitivity between 270 and 290 K is attributed to difference in the rate constant of DMS oxidation through the OH-addition channel. Did the authors test this statement by running the models with the same rate constant for this reaction?
I do not understand the authors reply – which rate constants are compared at 298 K and 1 atm? The authors should address the question in the manuscript text also.

Reply from the authors:
That's a great question, we didn't do that but as we use a consistent inorganic chemistry and we checked that things like OH are very similar between the model runs we felt we could safely assess this by just comparing the rate constants at 298 K and 1 atm. These are inconsistent across mechanisms hence we attribute this as the initiator of the divergence.

We apologise for the poor and confusing reply. The text here refers to the divergence shown in Figure 3 between the different UKCA DMS schemes; the ST (StratTrop) family (ST and ST~CS2) and the CS2 family (CS2, CS2-UPD-DMS, CS2-HPMTF). We infer (from inspection of the reaction fluxes and mechanism) that the cause of the difference in temperature dependence of the MSA profile in

Figure 3b (the gradient of d[MSA]/dT) is the difference in kinetics used for the DMS+OH addition channel. The expressions are given in Table 2 for the CS2 family of schemes and in S1.4.1 for the ST family of schemes and are visualised in Figure R1 to clearly have different gradients.

[Figure]

*Figure R1: A comparison of k DMS+OH addition channel as a function of temperature in the CS2 schemes and ST schemes shown in Figure 3.*

We have modified the text to make the point clearer that we are inferring it's the difference in the two expressions for the rate constants for the DMS+OH addition channel that drive the difference between MSA-Temperature gradients seen in the CS2 and ST families.

"We attribute this to differences in the rate constant of DMS oxidation through the OH-addition channel (see **Table 2** and **S1.4.1**) used in the UKCA ST schemes and the UKCA CS2 schemes. The expression used in the ST family of schemes (the provenance of which is Pham et al. (1995), see S1.4.1) has a much shallower gradient with temperature than the expression used in the CS2 family of schemes (which is based on the latest IUPAC recommendation)."